



# Continental and marine source regions contributing to the outflow of the Asian summer monsoon anticyclone during the PHILEAS campaign in summer 2023

Bärbel Vogel[1], Valentin Lauther[2], Franziska Köllner[3,4], Fatih Ekinci[3,4,5], Christian Rolf[1], Johannes Strobel[2], Ronja van Luijt[2], Michael C. Volk[2], Stephan Borrmann[3,4], Antonis Dragoneas[4], Oliver Eppers[4], Sergej Molleker[4], Peter Hoor[3], Linda Ort[4], Franziska Weyland[3], Andreas Zahn[6], Jan Clemens[1], Gebhard Günther[1], Oleh Kachula[1], Rolf Müller[1], Felix Ploeger[1,2], and Martin Riese[1]

[1]Institute of Climate and Energy Systems (ICE-4), Forschungszentrum Jülich, Jülich, Germany
[2]Institute for Atmospheric and Environmental Research, University of Wuppertal, Wuppertal, Germany
[3]Johannes Gutenberg University of Mainz, Institute for Atmospheric Physics, Mainz, Germany
[4]Max Planck Institute for Chemistry, Mainz, Germany
[5]Technical University of Darmstadt, Institute for Applied Geosciences (Environmental Mineralogy), Darmstadt, Germany
[6]Institute of Meteorology and Climate Research, Karlsruhe Institute of Technology, Karlsruhe, Germany

**Correspondence:** B. Vogel (b.vogel@fz-juelich.de)

**Abstract.**

During the PHILEAS aircraft campaign, which probed the export of air from the Asian summer monsoon anticyclone (ASMA), observations were carried out from Germany and Alaska during August and September 2023. The flights from

Germany sampled the western part of the ASMA and its western outflow, whereas filaments that had separated at its eastern flank and were transported across the Pacific were observed from Alaska. The aircraft measurements were combined with Lagrangian transport simulations. Our results show that air masses within the ASMA and its outflow are characterized by a mixture of different continental and marine sources. In the western part of the ASMA and its western outflow, enhanced pollutants and greenhouse gases such as $CH_2Cl_2$, aerosol, $CH_4$, and $H_2O$ were detected, indicating sources in South Asia.

In filaments separated at the ASMA's eastern flank, additional marine air masses from the western Pacific, characterized by enhanced $CH_2Br_2$ and low $O_3$, were found. Thus, air masses from two different surface regions and with different chemical compositions are mixed in filaments separated at the ASMA's eastern flank. Our findings show that the chemical composition of the ASMA's outflow is highly variable and depends on altitude, surface emissions, mixing of air masses from different source regions, and the interplay between the ASMA and tropical cyclones. The Asian summer monsoon is an important pathway for

transporting short-lived ozone-depleting and radiatively active substances into the stratosphere.

## 1 Introduction

During boreal summer, the Asian summer monsoon forms a seasonally persistent, zonally confined circulation system that efficiently transports climate-relevant emissions (e.g. greenhouse gases, ozone-depleting substances, aerosol particles, and their




gas-phase precursors) from near-surface levels to higher altitudes, specifically into the upper troposphere and lower stratosphere
(UTLS) (e.g. Mason and Anderson, 1963; Randel and Park, 2006; Park et al., 2007; Vogel et al., 2023). Between June and
September, deep convection over South Asia, particularly over the northern Indian subcontinent and China, is associated with
the formation of the Asian summer monsoon anticyclone (ASMA), which extends from northeastern Africa to the western
Pacific at UTLS levels (e.g. Park et al., 2007; Vogel et al., 2019). The strong anticyclonic circulation at the edge of the ASMA
acts as an effective transport barrier (e.g. Ploeger et al., 2015; Kachula et al., 2025), leading to the confinement of tropospheric
trace gases and aerosols within the anticyclone (e.g. Rosenlof et al., 1997; Li et al., 2005; Vernier et al., 2015; Chirkov et al.,
2016; Santee et al., 2017). However, filaments and sometimes smaller anticyclonic systems (eddies) are frequently shed from
the ASMA during boreal summer – particularly at its northeastern flank (e.g. over northeastern China, Korea, and Japan). These
structures are transported eastward along the subtropical jet stream towards North America (e.g. Alaska), and subsequently can
reach the Atlantic Ocean and Europe (e.g. Dethof et al., 1999; Garny and Randel, 2013; Ungermann et al., 2016; Clemens
et al., 2022). Likewise, air masses can be transported from the ASMA at its western flank (e.g. over northern Africa and the
Mediterranean region) westwards into the the tropical tropopause layer. Finally, these young air masses from the ASMA can
potentially be transported into the extratropical lower stratosphere and irreversibly mixed with the surrounding stratospheric
air (Vogel et al., 2016). This process affects the chemical composition and radiative balance of the northern extratropical UTLS
(Riese et al., 2012; Adcock et al., 2021). Consequently, during the Asian summer monsoon season, the northern extratropical
lower stratosphere is flooded – sometimes referred to as "flushed" (Hegglin and Shepherd, 2007; Müller et al., 2016) – with
moist, polluted air originating from South Asia, which is one of the most polluted and densely populated regions of the world.
This results in enhanced concentrations of greenhouse gases (e.g. $H_2O$, $CH_4$, $SF_6$), pollutants (e.g. CO, PAN, aerosols),
and short-lived ozone-depleting substances (e.g. $CH_2Cl_2$, $CHCl_3$), particularly those of anthropogenic origin in the northern
extratropical UTLS (e.g. Ploeger et al., 2013; Vogel et al., 2014, 2016; Rolf et al., 2018; Wetzel et al., 2021; Lauther et al.,
2022; Fadnavis et al., 2024; Graßl et al., 2024; Riese et al., 2025).

In the northwestern Pacific region, tropical cyclones (in particular typhoons) occur throughout the year, with peak activity
from July to October (Emanuel, 2003; Matsuura et al., 2003), coinciding with the peak phase of the ASMA. The interplay
between the spatial position of the ASMA and tropical cyclones plays a key role in controlling horizontal transport in the UTLS
of marine air uplifted in the western Pacific, in particular if tropical cyclones occur at the edge of the ASMA. Measurements
show that ozone-poor and aerosol-poor marine air, uplifted by tropical cyclones, can be injected into the anticyclonic flow of
the ASMA (e.g. Vogel et al., 2014; Li et al., 2017, 2020; Hanumanthu et al., 2020; Li et al., 2021; Clemens et al., 2024b).
This process can lead to reduced ozone in the UTLS over South Asia (Li et al., 2017, 2023). Further, wet marine air can
be uplifted into the UTLS by typhoons and enhance water vapour in this region, however dehydration processes can also
decrease water vapour in air masses uplifted by typhoons (Li et al., 2020, 2023). Hence, the rapid uplift by tropical cyclones
generally transports air of marine origin to altitudes of the ASMA, but upward transport of polluted air masses during landfall
of tropical cyclones also occurs. Further, subsequent filaments separated from the ASMA can transport boundary layer air from
the western Pacific into the northern extratropical UTLS (e.g. towards North America and Europe) (Vogel et al., 2014).



Several balloon campaigns have been conducted over the past decades across the Indian subcontinent and China, aiming to measure trace gases – particularly ozone, water vapour, and aerosol backscatter – within the ASMA (e.g. Bian et al., 2020; Brunamonti et al., 2018; Vernier et al., 2018; Fadnavis et al., 2023). As part of the StratoClim project, an aircraft campaign was carried out in Kathmandu (Nepal) during summer 2017, providing observations up to 20 km altitude and probing the top of the ASMA (e.g. Höpfner et al., 2019; Adcock et al., 2021; Vogel et al., 2024; Stroh and StratoClim-Team, 2025). Despite these unique measurements, further comprehensive in-situ data sets within the ASMA in other years remain lacking.

The eastward outflow of the ASMA over Europe and the North Atlantic Ocean was observed during two aircraft campaigns: the "Transport and Composition in the Upper Troposphere and Lowermost Stratosphere" (TACTS) campaign, conducted jointly with the "Earth System Model Validation" (ESMVal) experiment in August and September 2012 (e.g. Vogel et al., 2014, 2016; Müller et al., 2016; Gottschaldt et al., 2017; Rolf et al., 2018), and the "Wave-driven ISentropic Exchange" (WISE) campaign in autumn 2017 (e.g. Wetzel et al., 2021; Rotermund et al., 2021; Lauther et al., 2022). More recently, scientific efforts have focused on the eastern edge of the ASMA and its outflow towards the Northern Pacific, including the "Asian Summer Monsoon Chemical and CLimate Impact" (ACCLIP) project, conducted from South Korea in July and August 2022 (Pan et al., 2024, 2025; Smith et al., 2025), and the "Probing High Latitude Export of Air from the Asian Summer Monsoon" (PHILEAS) campaign, conducted from Oberpfaffenhofen, Germany, and Anchorage, Alaska, during summer and early autumn 2023 (Riese et al., 2025; Jesswein et al., 2025).

Here, we combine PHILEAS aircraft measurements with global three-dimensional simulations using the Chemical Lagrangian Model of the Stratosphere (CLaMS) (Pommrich et al., 2014; Vogel et al., 2015; Ploeger et al., 2021, and references therein), which includes model tracers of surface origin, as well as CLaMS back-trajectory calculations initiated along all aircraft flight tracks. With CLaMS, we identify the origin of air masses and their transport pathways and transport times to the location of the measurements. During PHILEAS, flights from Oberpfaffenhofen reached the western part of the ASMA over the Eastern Mediterranean, Israel, and Jordan. Flights from Anchorage focused on investigating outflow at the eastern flank of the ASMA. We will discuss three PHILEAS flights as case studies in more detail: measuring the ASMAS's western part, its western outflow and its eastern outflow. Further, we show that in addition to polluted air from South Asia also marine air from tropical cyclones in the western Pacific contributes to the chemical composition at the edge of the ASMA and its outflow.

## 2 Measurements during PHILEAS 2023

As part of the PHILEAS project, a measurement campaign using the German HALO (High Altitude and Long Range) research aircraft was conducted from Oberpfaffenhofen (Germany) and Anchorage (Alaska) during summer 2023 (Fig. 1) to investigate trace gas and aerosol characteristics of air affected by the Asian summer monsoon (Riese et al., 2025). Maximum flight altitudes of up to ∼14.5 km (approximately 410 K potential temperature) were reached. Flights from Oberpfaffenhofen (F02–F06, F20) targeted the western part of the ASMA over the Eastern Mediterranean, Israel, and Jordan, as well as the region north of Oberpfaffenhofen up to Spitsbergen to sample the lower northern extratropical background stratosphere. Flights





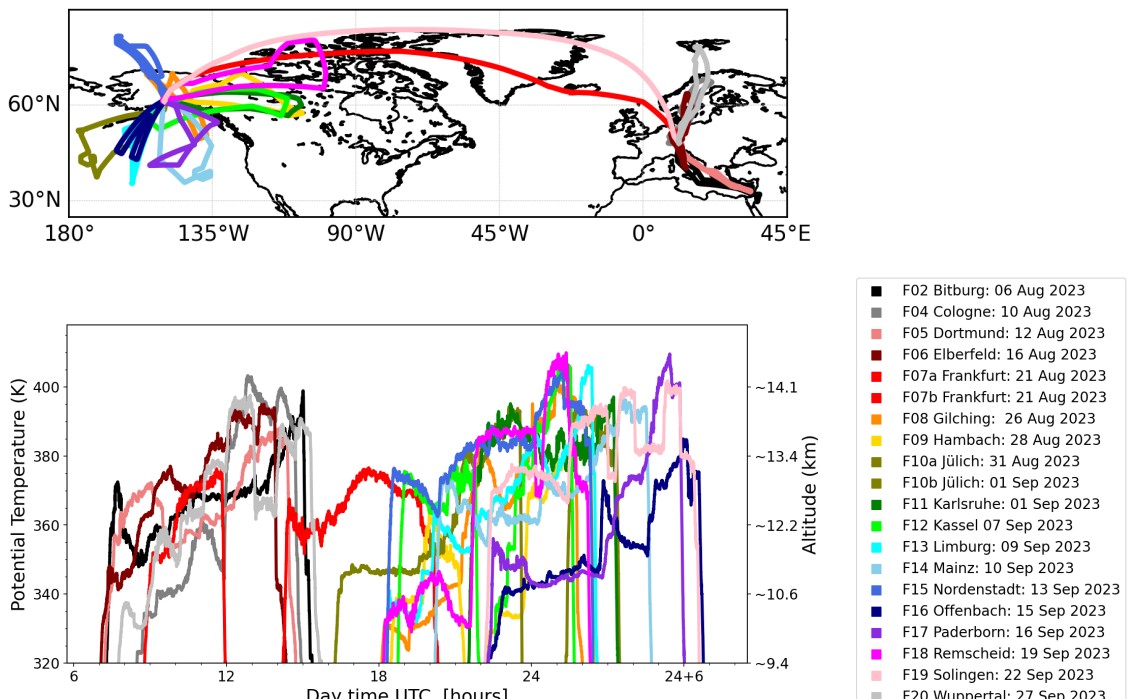

**Figure 1.** Regional map showing all HALO aircraft flight tracks over the Northern Hemisphere during the PHILEAS campaign 2023 (top). A total of 20 scientific flights (F02–F20; excluding the Electromagnetic Compatibility (EMC) flight F01) were conducted between 6 August and 27 September 2023, departing from Oberpfaffenhofen (Germany) and Anchorage (Alaska). Maximum altitudes up to ∼14.5 km (∼410 K potential temperature) were reached. Research flights from Alaska were conducted during local daytime, which corresponds to nighttime in Coordinated Universal Time (UTC) (Alaska local time = UTC - 7 h, Germany local time = UTC + 2 h considering summer time).

from Anchorage (F08–F18) focused on investigating the outflow at the eastern flank of the ASMA. Transfer flights between Oberpfaffenhofen and Anchorage (F07a, F07b, F19) are also included in the data set. The PHILEAS measurements provide a valuable data basis to characterise key processes governing the transport of particles and trace gases from South Asia – one of the most polluted regions of the world – into the northern UTLS. Measurements from the following in situ instruments on

board the HALO aircraft are used in our study:

– The chemical composition of aerosol particles (larger than ∼ 100 nm) was detected by the ERICA (ERC Instrument for the Chemical composition of Aerosols) Aerosol Mass Spectrometer (ERICA-AMS). It is based on thermal desorption with subsequent electron-impact ionisation, providing in-situ real-time mass concentrations of particulate nitrate, sulphate, and ammonium (Appel et al., 2022; Hünig et al., 2022). In addition, single-particle mass spectra were obtained

from the ERICA-LAMS part, which uses laser ablation and ionization mass spectrometry. With ERICA-LAMS, particle



fractions of certain particle types were analysed that are typically associated with marine-influenced air masses. Particle types were classified into three main categories based on characteristic ion marker peaks. Sodium chloride (Na/Cl) was identified by peaks at m/z +23, –35, and –37 ($Na^+$, $Cl^-$). Methanesulfonic acid (MSA) was identified by a peak at m/z –95 ($CH3SO_3^-$), and trimethylamine (TMA) by peaks at m/z +58 and +59 ($C3H8N^+$ and $C3H9N^+$) (Angelino et al., 2001; Rehbein et al., 2011; Healy et al., 2015; Köllner et al., 2017; Willis et al., 2017).

– Ozone ($O_3$) was measured with FAIRO, a lightweight (14.5 kg) instrument with high accuracy (2%) and high time resolution (10 Hz) developed for the HALO aircraft. FAIRO combines a dual-beam UV photometer with a UV-LED as a light source and a dry chemiluminescence detector (Zahn et al., 2012).

– Water vapour ($H_2O$) measurements were obtained from the Fast In situ Stratospheric Hygrometer (FISH), which is based on the Lyman-$\alpha$ photofragment fluorescence technique (Zöger et al., 1999). The FISH inlet was mounted facing forward to measure total water which is the sum of gas-phase water and water in ice particles. A correction procedure is applied to calculate gas-phase water from FISH measurements in clouds (for more details, see e.g. Meyer et al., 2015). The uncertainty for the water measurements by FISH during PHILEAS is 5% plus an offset of 0.8 ppmv. The relative humidity with respect to ice ($RH_i$) – a measure for potential cloud formation – is derived from $H_2O_{gas}$ using $RH_i = 100 \times H_2O_{gas} / H_2O_{sat,ice}$, where $H_2O_{sat,ice}$ is the ice saturation mixing ratio as a function of temperature (Murphy and Koop, 2005; Krämer et al., 2016).

– Dichloromethane ($CH_2Cl_2$) and dibromomethane ($CH_2Br_2$) were measured by HAGAR-V on board of HALO. HAGAR-V comprises a two-channel gas chromatograph (GC) with electron capture detection (ECD) as well as a non-dispersive infrared absorption module for the detection of $CO_2$. In addition, a mass spectrometer (MS) coupled to two GC channels by a two-position valve to alternately use the detector and thereby double the time resolution of the wide range of measured species, including $CH_2Cl_2$ and $CH_2Br_2$, is integrated (Lauther et al., 2022).

– Methane ($CH_4$) measurements are used from UMAQS, the University of Mainz Airborne Quantum Cascade Laser (QCL) Spectrometer (e.g. Müller et al., 2015; Kunkel et al., 2019). Direct absorption of an infrared QCL laser at 2989 $cm^{-1}$ for $CH_4$ at a constant cell pressure of 40 Torr is used to determine its concentration, which is converted to mixing ratio using measured cell pressure and temperature. The instrument does in-flight calibrations against secondary gas standards of compressed dried ambient air, which are compared to standards from NOAA prior and after the campaign. With this setup, a total uncertainty (1 sigma) for the 1 Hz data set of 0.7 ppbv for $CH_4$ under in-flight conditions on straight legs was achieved.

## 3  Lagrangian transport simulations

The Chemical Lagrangian Model of the Stratosphere (CLaMS) (McKenna et al., 2002b, a; Pommrich et al., 2014, and references therein) was developed to study transport and chemical processes throughout the upper troposphere and stratosphere.





In CLaMS, the diabatic approach is used to calculate vertical velocities at these atmospheric altitudes. Specifically, potential temperature is used as the vertical coordinate at pressures below approximately 300 hPa, i.e. in the upper troposphere and stratosphere. At higher pressures (for $p/p_{\text{surface}} > 0.3$), a pressure-based, orography-following hybrid coordinate (expressed in units of K) is applied (Pommrich et al., 2014). At potential temperature levels above about 300 hPa, vertical velocity is determined solely by the total diabatic heating rate from ERA5 reanalysis (Pommrich et al., 2014; Ploeger et al., 2021). Total diabatic heating rates include clear-sky radiative heating, cloud radiation, latent heat release, as well as turbulent and diffusive heat transport for the upper troposphere and stratosphere.

In this study, high-resolution ERA5 reanalysis data (Hersbach et al., 2020) are used with 137 vertical levels up to 0.01 hPa, a horizontal resolution of $\sim 31$ km (according to a spectral trunction of $T_L$ 639) and an hourly time resolution provided by the European Centre for Medium-Range Weather Forecasts (ECMWF). We retrieved the ERA5 data on a regular $0.3° \times 0.3°$ horizontal grid. The high-resolution ERA5 reanalysis was already used for pure CLaMS trajectory calculation (Li et al., 2020; Vogel et al., 2023, 2024; Clemens et al., 2024b, a) and is here used for global three-dimensional CLaMS simulations. Technical improvements were implemented to accelerate the reading of ERA5 reanalysis data (here used in netCDF format). Furthermore, the storage of the ERA5 reanalysis data was changed from distributed memory to shared memory using the Message Passing Interface (MPI).

The upward transport and convection in CLaMS (in both trajectory calculations and three-dimensional simulations) depend on the underlying reanalysis data. In ERA5, the representation of convection and tropical cyclones (e.g. typhoons) is substantially improved compared to its predecessor, ERA-Interim (e.g. Hoffmann et al., 2019; Li et al., 2020; Malakar et al., 2020; Clemens et al., 2024b). In our study, no additional parametrisation for convection (Konopka et al., 2019, 2022) is applied in the CLaMS simulations; only the convection resolved in the ERA5 reanalysis is used. Thus, small-scale convection that is unresolved in ERA5 is not considered in our study, therefore the fractions of the surface–origin tracers are a lower limit.

### 3.1 Lagrangian three-dimensional simulations including surface–origin tracer

Three-dimensional CLaMS simulations that include irreversible mixing (applied here every 24 h) (e.g. Konopka et al., 2007) provide a very good representation of observed tracer gradients in the UTLS, particularly near the tropopause and at the edge of the Asian summer monsoon anticyclone (e.g. Vogel et al., 2015, 2019; Ploeger et al., 2017, 2024).

Here, we use CLaMS simulations with 32 different regional model tracers of surface origin (hereafter referred to as 'surface–origin tracers') that together cover the entire Earth's surface (Fig. 2). These surface–origin tracers are released into the model boundary layer (approximately 2–3 km above the surface, accounting for orography) every 24 h and are subsequently transported – by advection and mixing – into the free atmosphere over the course of the simulation. In addition, a surface–origin tracer representing the global model boundary layer (MBL), corresponding to the sum of all regional surface–origin tracers shown in Fig. 2, is included.

The simulation starts on 1 May 2023, during the pre-monsoon period, and covers the entire Asian summer monsoon season of 2023. Surface–origin tracers serve as a valuable diagnostic tool for identifying the surface source regions of air masses. In





addition, the global MBL tracer (i.e. the sum of all regional surface–origin tracers) provides an estimate of the age of the air: low MBL fractions indicate a high fraction of aged air masses (i.e. air older than 1 May 2023), such as stratospheric background air.

Previous model studies using CLaMS surface–origin tracers (albeit with a slightly different definition of tracer regions) indicate that young air masses originating from South Asia are largely confined within the ASMA. In contrast, air masses from adjacent regions – particularly from Southeast Asia and the western Pacific – tend to be located near the edge of the ASMA (e.g. Vogel et al., 2015, 2019; Becker et al., 2025).

In this study, we use the sum of the following surface–origin tracers to represent air from the ASMA, referred to as the 'South Asia' tracer: Northern Indian Subcontinent (NIN), Indian Subcontinent (IND), Tibetan Plateau (TIB), Eastern China (ECH), Bay of Bengal (BoB), Northern Indian Ocean (NIO), as well as the Near East (Neast) and Northern Africa (NAF), with the latter two contributing only in small fractions. Air masses from the western Pacific, that is the sum of the surface–origin tracers of Southeast Asia (SEA), Warm Pool (Wpool), Tropical Western Pacific (TWP) and Northern Western Pacific (NWP) (Fig. 2) also play a role in the chemical composition of the Asian Summer Monsoon Anticyclone (ASMA) (e.g. Li et al., 2017, 2023; Hanumanthu et al., 2020; Vogel et al., 2023; Clemens et al., 2024b) and are frequently observed at its edge (e.g. Li et al., 2017; Vogel et al., 2019), partly due to uplift associated with tropical cyclones. These surface–origin tracers – especially the South Asia tracer (excluding contributions from the Near East and Northern Africa) – were used for flight planning during the PHILEAS campaign (Riese et al., 2025) to identify the ASMA, as well as filaments and anticyclonic structures (eddies) detached from it, where simulations with ECMWF forecast and analysis products were used for flight planning. We demonstrate that the South Asia tracer serves as a reliable proxy for polluted air originating from the Asian summer monsoon region. Additionally, the Western Pacific tracer is a useful indicator of air uplifted into the UTLS by tropical cyclones in the western Pacific, and more generally, it serves as a marker for air masses of marine origin.

### 3.2 CLaMS back-trajectory calculations

CLaMS diabatic back-trajectories were initiated along the entire flight paths (every 1 second) of all 20 research flights conducted during the PHILEAS campaign. On average, the research flights lasted 6 to 8 hours, resulting in approximately 23,000 to 33,000 trajectories calculated per flight, depending on flight duration. Trajectories are considered to terminate in the model boundary layer when they first descend to below approximately 2–3 km above the surface, accounting for orography (i.e., when the vertical hybrid pressure-potential-temperature coordinate, $\zeta$, satisfies $\zeta \leq 120\,\mathrm{K}$) (see details in, e.g., Vogel et al., 2023, 2024). The transport time of each air parcel is defined as the time difference between the start and end points of its back-trajectory in the model boundary layer. Trajectories not ending in the model boundary layer were calculated back to 1 May 2023; representing older air found in the UTLS region. Thus, the length of these trajectories span the same time period as the global three-dimensional CLaMS simulations that include surface–origin tracers (i.e., a quasi-forward calculation including irreversible mixing) and cover both the pre-monsoon and monsoon period. Along the CLaMS back-trajectories as well as along





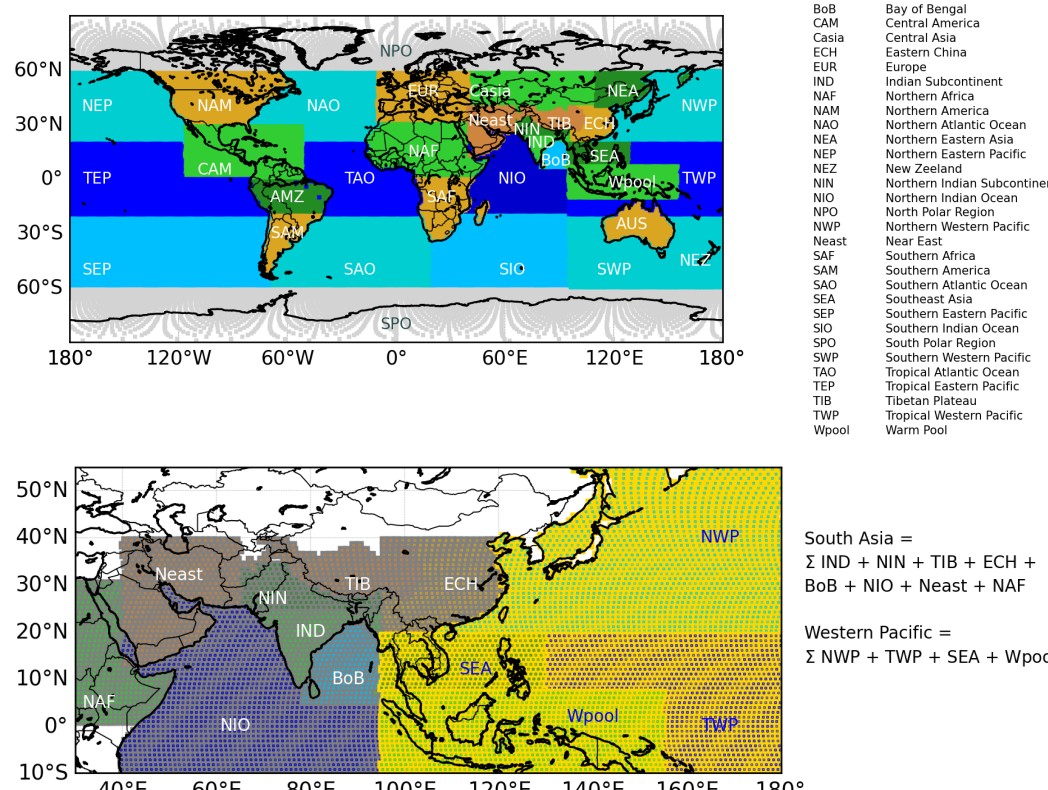

**Figure 2.** Geographical map showing the global distribution of surface–origin tracers implemented in CLaMS (top). The South Asia tracer is defined as the sum of several surface–origin tracers in the Asian summer monsoon region (TIB + IND + NIN + ECH + BoB + NIO + Neast + NAF), and serves as a good proxy for the Asian Summer Monsoon Anticyclone (bottom, grey background). The Western Pacific tracer (SEA + Wpool + TWP + NWP) is a good indicator of air uplifted into the UTLS by tropical cyclones in the western Pacific and, more generally, of air masses with marine origin (bottom, yellow background).



the flight tracks of all scientific PHILEAS flights the local tropopause heights (first and second) were calculated from ERA5
reanalysis data based on the WMO definition according to Hoffmann et al. (2022).

In addition, the temperature evolution along the back-trajectories is used to check if dehydration can possibly occur along
the trajectories. That is in particular important at the convective outflow of tropical cyclones (e.g. Li et al., 2020), but also
tropical convection can impact hydration and dehydration in the UTLS (e.g. Schoeberl et al., 2018; Ueyama et al., 2023). To
estimate dehydration, the minimum of the water vapour saturation mixing ratio with respect to ice ($H_2O_{sat,ice,min}$) along the
200 back-trajectories is calculated based on ERA5 temperatures and pressures. Dehydration is only relevant in a first approximation
for air masses that are transported from the troposphere into the lower stratosphere. Therefore, tropopause heights along the
trajectories are used to check whether the trajectories originate from the troposphere before reaching $H_2O_{sat,ice,min}$ and remain
in the stratosphere afterwards.

If dehydration can be described with this simple Lagrangian model by minimum saturation along the backward trajectories,
tropopause heights along the trajectories are used to check whether the trajectories originate from the troposphere before
reaching $H_2O_{sat,ice,min}$ and remain in the stratosphere afterwards. Therefore, air parcels along the flight track are marked as
dehydrated when the following criteria are fulfilled: trajectory (at least 80% of the time) is below tropopause before reaching
$H_2O_{sat,ice,min}$ and above afterwards, in addition FISH $H_2O$ measurements (total water) should be greater equal $H_2O_{sat,ice,min}$
and to exclude measurements inside of clouds, FISH $RH_i$ has to be lower than 90%.

## 4  Results

### 4.1  Air mass origin and impact of tropical cyclones

In this study, we focus on the UTLS, and therefore only consider back-trajectories of aircraft measurements taken above 340 K
potential temperature. To determine the origin of the air masses sampled during the PHILEAS aircraft campaign, it is crucial
to identify the location where the back-trajectories intersect the model boundary layer. Figure 3 (top) shows the frequency
distribution of locations where air parcels were traced back to the model boundary layer for all CLaMS back-trajectories
initiated above 340 K across all 20 research flights (F02-F20).

Above 340 K, most back-trajectories originate from two main regions (Fig. 3, middle): (1) South Asia – particularly the
Indian Subcontinent, the Arabian Sea, the Bay of Bengal, the Tibetan Plateau, China, and Mainland Southeast Asia – and (2)
the Western Pacific, including the eastern coast of China and the regions around Taiwan and the Philippines. The frequency
distribution of locations where air parcels were traced back to the model boundary layer motivated our definition of the South
Asia tracer and Western Pacific tracer introduced in Sect. 3.1.

When focusing on measurements conducted at altitudes of the ASMA and above the maximum level of convective outflow
(Brunamonti et al., 2018; Gettelman and de Forster, 2002) i.e. here measurements above about 360 K (Fig. 3, bottom), the
regions with the highest frequency distribution of air mass origins are the Arabian Sea, the Bay of Bengal, the north eastern
part of the Indian Subcontinent and the western Pacific. These regions all show frequent occurrence of strong convection.





Particularly in the western Pacific, frequent occurrence of strong tropical cyclones is found impacting the PHILEAS measurements. Figure 4 shows the frequency distribution of locations where CLaMS back-trajectories are traced back to the model boundary layer zoomed to the western Pacific and overlayed by all cyclone tracks which occurred in the western Pacific between 6 June and 8 September 2023 and possibly have an impact on the PHILEAS measurements (provided by the Japan Meteorological Agency; (https://www.jma.go.jp/jma/jma-eng/jma-center/rsmc-hp-pub-eg/besttrack.html). The cyclone category is indicated along each cyclone track: tropical depression, tropical storm, severe tropical storm, typhoon and extra-tropical cyclone.

A strong local coincidence between the frequency distribution of air mass origins and cyclone tracks is found for cyclones (B) Talim, (C) Doksuri, (G) Saola and (I) Haikui north of the Philippines; the latter three were all categorised as a typhoon in this region (Fig. 4). Furthermore, a strong spatial overlap is observed for tropical cyclones (D) Khanun during its transition from a typhoon to a severe tropical storm off the eastern coast of China, and later during its landfall in the region of Japan, Korea, and the Chinese coast. Upon landfall, tropical cyclones have the potential to rapidly uplift polluted boundary layer air from coastal regions into the UTLS. Further details are discussed for selected PHILEAS flights in Sect. 4.3.

### 4.2 Tracer–tracer relations

Tracer–tracer relations of different measured trace gases are commonly used to diagnose mixing between air masses of different origin, such as between tropospheric and stratospheric air masses (e.g. Hoor et al., 2002; Hegglin and Shepherd, 2007; Pan et al., 2007). In this study, tracer–tracer relations based on in situ measurements are combined with surface–origin tracers from three–dimensional global CLaMS simulations (e.g. Vogel et al., 2011; Lauther et al., 2022). The CLaMS surface–origin tracers are interpolated in space and time along the flight paths of all PHILEAS research flights (F02–F20). Only flight segments above 340 K are considered here in tracer–tracer relations relations to remove the impact of the lower troposphere.

The aim of our further analysis is to demonstrate that air masses originating from South Asia exhibit a different chemical composition compared to those from the western Pacific. Mixing between air from these two source regions influences the chemical composition of air within the Asian Summer Monsoon Anticyclone (ASMA) itself and consequently its outflow to the west and to the east. In addition, mixing with older stratospheric air must be taken into account.

### 4.2.1 $CH_4$–$CH_2Cl_2$ relations indicating pollution sources in South Asia

South Asia is a substantial source of methane ($CH_4$) – an important greenhouse gas, primarily due to elevated emissions from rice paddies (e.g. Park et al., 2004; Schuck et al., 2010; Tao et al., 2024). These emissions lead to a statistically significant increase in $CH_4$ in the ASMA and subsequently in the northern extratropical UTLS during summer and autumn (e.g. Baker et al., 2012; Rolf et al., 2018; Zhu et al., 2025). Air masses with $CH_4$ mixing ratios exceeding a certain threshold value for $CH_4$ were used as a proxy of air from the Asian monsoon region. In studies related to the PHILEAS campaign different $CH_4$ threshold values were introduced: 1850 ppbv in Riese et al. (2025) referring to Rolf et al. (2018) and 1920 ppbv in a recent study by Köllner et al. (2025). Because only flight segments above 340 K are considered here, enhanced $CH_4$ measured in



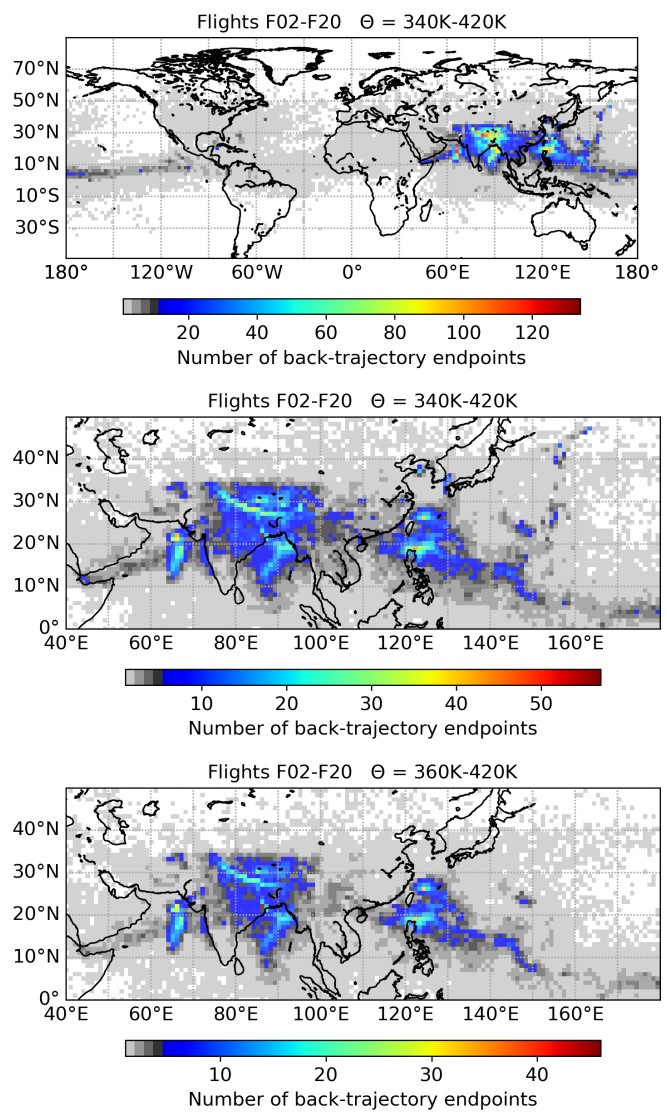

**Figure 3.** Frequency distribution of locations where air parcels along the flight tracks of all PHILEAS research flights (F02-F20) were traced back to the model boundary layer using CLaMS back-trajectory calculations. Frequency distributions are shown for measurements above 340 K in a global view (top) as well as zoomed in on South Asia and the western Pacific. To highlight sources in Asia reaching altitudes of the ASMA an additional frequency distribution for measurements above 360 K is shown (bottom).

the lower troposphere over Europe and North America and in particular measured during take-off and landing of the research
aircraft are not taken into account in our analysis.



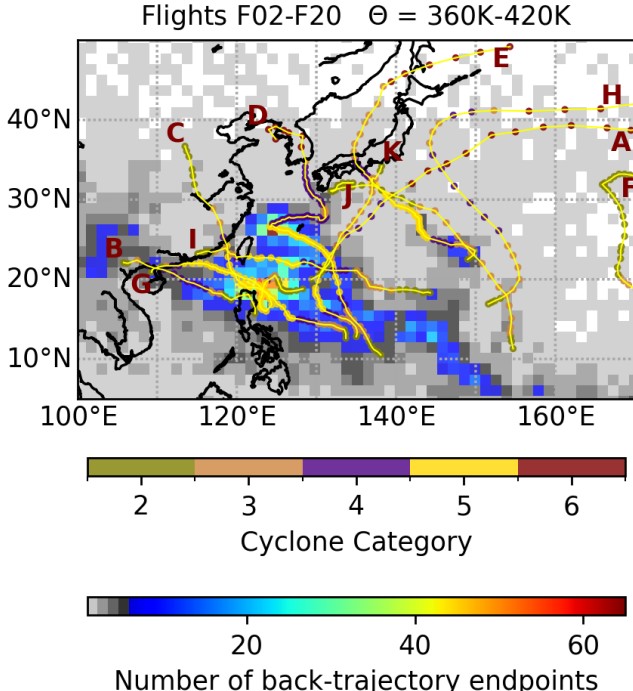

**Figure 4.** Frequency distribution of locations where air parcels were traced back to the model boundary layer using CLaMS back-trajectory calculations (for aircraft measurements taken above potential temperature levels of 360 K), overlaid with cyclone tracks in the western Pacific. Cyclone categories are indicated as follows: Tropical Depression (2), Tropical Storm (3), Severe Tropical Storm (4), Typhoon (5), Extratropical Cyclone (6). The following cyclones are included: (A) Guchol (6–16 June 2023), (B) Talim (13–18 July 2023), (C) Doksuri (20–30 July 2023), (D) Khanun (26 July – 11 August 2023), (E) Lan (7–18 August 2023), (F) Dora (12–22 August 2023), (G) Saola (22 August – 3 September 2023), (H) Damrey (23–30 August 2023), (I) Haikui (27 August – 6 September 2023) (J) Kirogi (29 August – 6 September 2023) and (K) Yun-Yeung (4–8 September 2023). The end of each cyclone track is marked by the corresponding capital letter.

Dichloromethane ($CH_2Cl_2$), a chlorine-containing very short-lived halogenated substance (VSLS), originates mainly from anthropogenic sources – in particular from South Asia (Feng et al., 2018; Say et al., 2019). An average annual increase of 13% in emissions of the industrially produced $CH_2Cl_2$ was reported for the time period between 2011 and 2019 – primarily from eastern China (An et al., 2021). Over the Indian subcontinent $CH_2Cl_2$ mixing ratios up to ∼140 pptv were measured during the StratoClim aircraft campaign in 2017, a factor of more than two more than reported previously at the tropical tropopause (∼50 pptv) (Adcock et al., 2021). During the ACCLIP aircraft campaign in 2022 record-breaking mixing ratios up to ∼ 600 pptv $CH_2Cl_2$ were measured at tropopause altitudes at the eastern edge of the ASMA (over Korea and Japan) (Pan et al., 2024). Export of air from the ASMA to the northern extratropical UTLS leads to enhanced $CH_2Cl_2$ mixing ratios over the Atlantic Ocean and North Europe (Lauther et al., 2022) as well as in the Arctic (Laube et al., 2025).





In this study, we use $CH_4$–$CH_2Cl_2$ relations as an indicator of air masses influenced by South Asian emissions. In the $CH_4$–$CH_2Cl_2$ relation from all scientific PHILEAS flights (F02–F20), three distinct branches (1, 2, 3) emerge with varying $CH_2Cl_2$ mixing ratios at high $CH_4$ levels (greater than 2000 ppbv): up to 300 pptv of $CH_2Cl_2$ (branch 1 in Fig. 5), around 150 pptv (branch 2 in Fig. 5), and nearly 100 pptv (branch 3 in Fig. 5). All three branches are associated with contributions of the global model boundary layer (MBL; released since 1 May 2023) tracer exceeding 60%, indicating young air masses and reflecting the short atmospheric lifetime of $CH_2Cl_2$ (~6 months). Branches 1 and 2 correspond to measurements around 350 K potential temperature, while branch 3 was observed at higher altitudes, around 370 K. Thus only branch 3 constitutes measurements at altitudes of the ASMA, demonstrating that measured $CH_2Cl_2$ mixing ratios much greater than $\approx$ 150 pptv are measured at altitudes below the main anticyclone (Jesswein et al., 2025; Riese et al., 2025).

Looking into the origin of the air masses using the South Asia and the Western Pacific tracer it turns out that branches 1 and 2 have intermediate contributions from South Asia (reddish colours; fractions greater than 45% ) mixed with lower fractions from the Western Pacific (~25-35%), except in the part of the relation in Fig. 5 (bottom left) denoted by 1b, where also substantial fractions from the Western Pacific (~40-50%) were found. In contrast, in branch 3 the fractions from South Asia are up to 70%, with minor contributions from the Western Pacific (tracer lower than 10%). High fractions of the Western Pacific tracer (up to 70%) are found for $CH_4$ mixing ratios of ~1950 ppbv and $CH_2Cl_2$ mixing ratios of ~50–60 pptv denoted as region 4 in Fig. 5. These $CH_2Cl_2$ mixing ratios correspond to the northern hemispheric background $CH_2Cl_2$ values from ground-based measurements (https://agage.eas.gatech.edu/data_archive/data_figures/monthly/pdf/CH2Cl2_mm.pdf; last access: 5 November 2025) and therefore region 4 indicates marine air from the western Pacific.

To demonstrate that other parts of the world, besides South Asia and the Western Pacific, have only a minor impact on the PHILEAS measurements, the $CH_4$–$CH_2Cl_2$ relation is shown for surface–origin tracers representing the northern background (only relevant for local $CH_4$ emissions observed during take-off and landing of the HALO aircraft) and the residual surface (mainly parts of the southern hemisphere that have no impact on the PHILEAS flights) in Appendix A1 (Fig. A1).

To demonstrate further details about the source regions of high $CH_2Cl_2$ mixing ratios, the $CH_4$–$CH_2Cl_2$ relation for all surface–origin tracers contributing to the South Asia and Western Pacific tracer are shown in Appendix A1 (Fig. A2). Our findings show that branch 1 is related to air mass origins in eastern China (ECH) and the Northern Western Pacific (NWP; in particular in region 1b), branch 2 to eastern China (ECH), Northern Indian Subcontinent (NIN) and the Tropical Western Pacific (TWP) and branch 3 to Indian Subcontinent (IND) and Bay of Bengal (BoB). The marine air from the Western Pacific is dominated by fractions from the Northern Western Pacific (NWP) and the Tropical Western Pacific (TWP).

At potential temperature levels of the ASMA (greater than 360 K), only enhanced values of $CH_4$ and $CH_2Cl_2$ are found in PHILEAS observation made from Oberpfaffenhofen in Germany (F02, F05), probing the western part and outflow of the ASMA. In the following, we will discuss three research flights (F02, F06, F08) as case studies focusing on potential temperature levels of the ASMA (greater than 360 K) to analyse the impact of the origin of the air masses on the chemical composition of the ASMA and its outflow as well as the impact of tropical cyclones at these levels of potential temperatures.





**Figure 5.** $CH_4$–$CH_2Cl_2$ relations for all scientific PHILEAS flights F02–F20, colour-coded by entire model boundary layer (MBL) and potential temperature (top) for measurements above 340 K. The MBL tracer serves as an indicator for young air (younger than 1 May 2023). The South Asia (TIB + IND + NIN + ECH + BoB + NIO + Neast + NAF) and Western Pacific (SEA + Wpool + TWP + NWP) tracer indicated different origins of air in different part of the $CH_4$–$CH_2Cl_2$ relation.



### 4.3 Case studies of selected flights

#### 4.3.1 Research flight F02 (6 August 2023): The western part of the anticyclone

Research flight F02 on 6 August 2023 was conducted from Oberpfaffenhofen, Germany, to the Mediterranean area and intruded into the western part of the ASMA located over the eastern Mediterranean region (Fig. 6, left). To indicate the edge of the ASMA, its horizontal boundary is calculated on isentropes using the Montgomery streamfunction based on the recently published method by Kachula et al. (2025). A belt of air from the western Pacific is found at the outer edge of the ASMA that was crossed by the HALO aircraft twice (Fig. 6, right). Further, at the turning point of the flight track above the Arabian

Peninsula, a filament of air from the western Pacific was reached caused by a meandering subtropical jet. Air masses measured in the western part of the ASMA and in its western outflow mainly originate from the Indian subcontinent (IND, NIN, BoB, TIB, NIO); the main contributions from the western Pacific are from Southeast Asia (SEA) (Fig. 7). The South Asia tracer and the Western Pacific tracer along the flight track show opposite variations indicating that both air masses are separated and not well mixed. This is caused by the transport barrier at the edge of the ASMA.

Parts of research flight F02 in particular in the western part of the ASMA are below the local tropopause (Fig. 8), thus characterise tropospheric air masses. In contrast to the end of the flight outside the ASMA over Europe, stratospheric air was measured. The time series of the surface–origin tracers interpolated along the flight track of research flight F02 are compared with measurements of different chemical trace gases namely $CH_2Cl_2$, $CH_4$, $CH_2Br_2$, $O_3$ and $H_2O$ including $RH_i$ as well as particulate nitrate as indicator for the Asian tropopause aerosol layer (ATAL) (e.g. Höpfner et al., 2019; Appel et al.,

2022; Köllner et al., 2025) and the number fraction of particulate TMA (Fig. 8). The number fraction refers to the number of particles containing TMA divided by the total number of particles measured within the 10 min time interval. For research flight F02 enhanced fractions of the South Asia tracer are are found simultaniously with enhanced mixing ratios of pollutants and greenhouse gases such as $CH_2Cl_2$, $CH_4$ and enhanced particulate nitrate and TMA (time interval 2).

Interval 2 (Fig. 8) is flanked by interval 1 and 3 – crossing the belt of air from the western Pacific at the edge of the ASMA –

here fractions of the Western Pacific tracer of up to 35% are simulated. Within these flight segments mixing ratios of pollutants and greenhouse gases such as $CH_2Cl_2$, $CH_4$ as well as mass concentrations of particulate nitrate are low or are decreasing (e.g. in interval 3). Marine sources are indicated by low ozone mixing ratios ($\lesssim 100$ ppbv) or enhanced dibromomethane $CH_2Br_2$ ($\gtrsim 0.7$–0.8 pptv) (Adcock et al., 2021) that has mostly natural oceanic sources in contrast to chlorinated VSLS such as $CH_2Cl_2$ that has mainly anthropogenic sources. In all three intervals 1–3, ozone mixing ratios are low and $CH_2Br_2$ mixing ratios are

enhanced indicating marine sources, however there is no strong difference between interval 2 compared to interval 1 and 3.

An increased fraction of TMA-containing particles during intervals 2 and 3 (Fig. 8, bottom) was observed. Gaseous TMA originates from both natural and anthropogenic sources, including emissions from marine biota, biomass burning, and animal husbandry (Gibb et al., 1999; Facchini et al., 2008; Ge et al., 2011). Once released into the atmosphere, TMA can participate in aerosol chemistry through gas-to-particle conversion and dissolution in cloud droplets. ERICA-LAMS single-particle spectra

further indicate that TMA was internally mixed with nitrate, ammonium, and sulfate (see Fig. B1 in Appendix B). These observations suggest, first, that TMA may contribute to particle formation and growth within the ASMA, alongside ammonium,



nitrate, and sulfate. Second, they point to South Asia and the Western Pacific as potential source regions, and/or to shared emission sources – such as agricultural activities in northern India – for TMA and other compounds like ammonia.

In general, the Asian summer monsoon is a source of water vapour in the UTLS caused by the relatively high cold point
tropopause in the ASMA. (Rosenlof et al., 1997; Ploeger et al., 2013; Rolf et al., 2018). Also during flight F02 the tropopause is at higher potential temperature during flight segments within the ASMA (corresponding to enhanced fractions of the South Asia tracer; Fig. 8). Research flight F02 is the only flight probing air from inside the western part of the ASMA, therefore relative high water vapour mixing ratios (up to ∼30 ppmv) in interval 2 (Fig. 8) are measured above 360 K potential temperature (for more details to $H_2O$ mixing ratios during PHILEAS see Fig. A3 in Appendix A2). In general, $H_2O$ is decreasing with
increasing potential temperature, however, there is also a lot of additional variability of FISH total water mixing ratios along the flight path of research flight F02. Low fractions of the relative humidity with respect to ice ($RH_i$) (<20%) indicate that cloud formation did not occur during the measurements in intervals 1–3, thus FISH total water mixing ratios correspond here to gas-phase $H_2O$ mixing ratios. FISH $H_2O$ in intervals 1 and 3, are different compared to water vapour in interval 2 at adjacent levels of potential temperature below the local tropopause. These findings yield a hint that air masses in intervals 1 and 3 or
parts of them have a different origin and transport pathway compared to air masses measured in interval 2.

Analysing the back-trajectories from Flight F02, main source regions are the Bay of Bengal, the Northern Indian Subcontinent and the Western Pacific (Fig. 9) in accordance with CLaMS simulations using surface–origin tracers (Fig. 8). Interval 2 is dominated by air from the Bay of Bengal and the Northern Indian Subcontinent, whereby air masses in interval 1 and 3 were traced back to the Western Pacific. The trajectory endpoints in the model boundary layer have a strong coincidence to the track
of tropical cyclone Doksuri in particular to the location where it reaches the grade of a typhoon. Our findings show that air masses from the western Pacific found in interval 1 and 3 are uplifted by tropical cyclone Doksuri.

The temporal evolution of the South Asia tracer at 370 K potential temperature (Fig. C1 in Appendix C) shows, that high fraction of the South Asia tracer were uplifted in the Bay of Bengal and adjoining continental regions to the UTLS between 1 and 3 August 2023 and subsequently were transported westwards with the anticyclonic flow of the ASMA to the Mediterranean
area. The temporal evolution of the Western Pacific tracer at 370 K potential temperature (Fig. C2 and C3 in Appendix C) shows the impact of tropical cyclone Doksuri uplifting high fractions of the Western Pacific tracer up to 370 K, thus to potential temperature levels of the UTLS. Subsequently, enhanced fractions of the Western Pacific tracer uplifted by tropical cyclone Doksuri (from 23 to 27 July) were transported anticyclonically around the ASMA's southern edge. Thereafter, at the western flank of the ASMA over the Atlantic Ocean these air masses were captured by the subtropical jet and were transported eastwards. Thus,
a belt of air with enhanced fractions of the Western Pacific tracer is forming between the northern edge of the ASMA and the subtropical jet that was measured during research flight F02 (Fig. 6).

Thus, simulations with CLaMS indicate that air masses at the edge of the ASMA have a different origin compared to air masses inside the anticyclone which is in agreement with signatures found in measurements of different chemical trace gases and in the occurrence of particles formed within the ATAL (here indicated by nitrate mass concentrations). Further the TMA
particle fractions found in aerosol particles in interval 2 and 3 indicate the influence from marine sources and/or sources in





regions within rural areas with livestock farming and biomass burning such as in regions around the Bay of Bengal (West Bengal or Bangladesh).

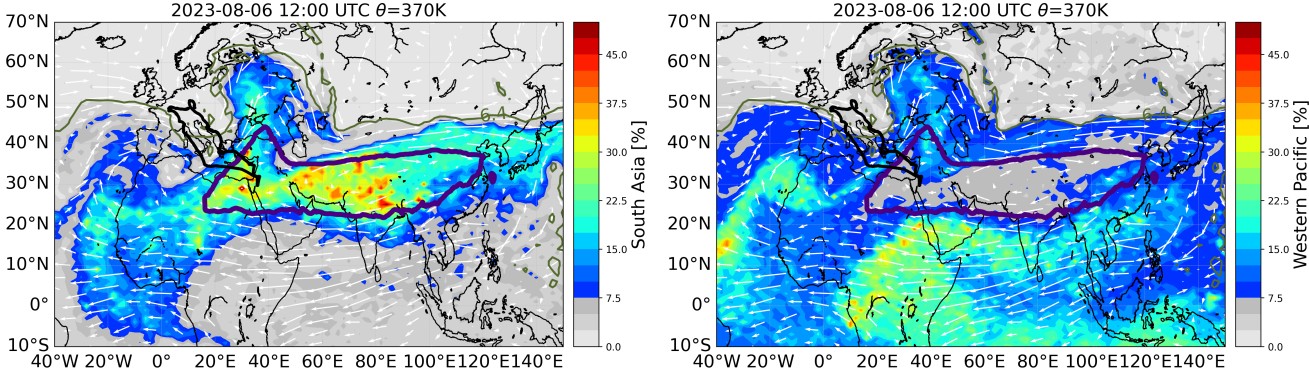

**Figure 6.** Research flight F02 on 6 August 2023 conducted from Oberpfaffenhofen, Germany, to the Mediterranean area intruding into the western part of the ASMA (South Asia surface–origin tracer at 370 K 12:00 UTC, left). A belt of air from the western Pacific is found around the outer edge of the ASMA (Western Pacific surface–origin tracer at 370 K 12:00 UTC, right). The surface–origin tracer distributions are based on a CLaMS simulation driven by ERA5. The calculated synoptic HALO flight track position at 12:00 UTC is indicated by the black line. To indicate the edge of the ASMA (indigo line), the boundary of the ASMA is calculated using the Montgomery streamfunction. An optimised background value gives the ASMA boundary (MSF = 357.3 $m^2 s^{-2}$) for 6 August 2023 12:00 at 370 K using ERA5 reanalysis data based on the method by Kachula et al. (2025). The climatological isentropic transport barrier (PV = 6.4 PVU) derived by Kunz et al. (2015) for the Northern Hemisphere at 370 K during summer indicates the barrier between the tropical tropopause layer and the extra-tropical lower stratosphere (olive line). Further, horizontal winds are indicated by white arrows.

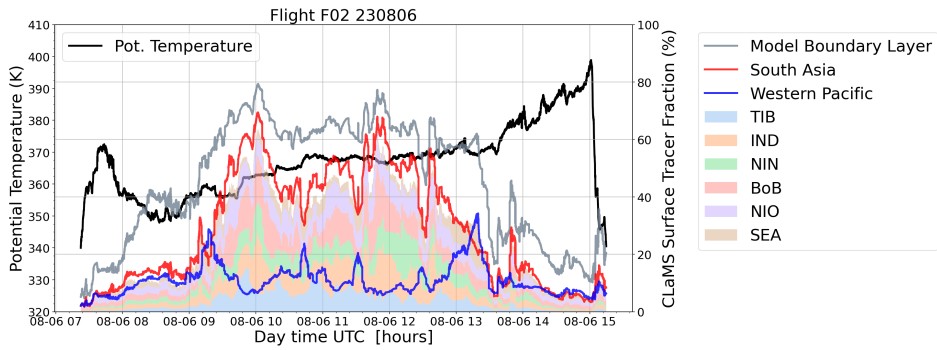

**Figure 7.** Surface–origin tracers for the model boundary layer, South Asia and the Western Pacific interpolated in space and time along the flight track of research flight F02 on 6 August 2023. In addition individual surface–origin tracers with fractions greater than 10% are shown (TIB, IND, NIN, BoB, NIO and SEA).






**Figure 8.** Surface–origin tracers interpolated along the flight track of research flight F02 on 6 August 2023, potential temperature, potential temperature at the tropopause as well as measurements of different chemical trace gases such as $CH_2Cl_2$, $CH_4$, $CH_2Br_2$, $O_3$, $H_2O$, relative humidity with respect to ice $RH_i$ as well as nitrate mass concentration and TMA fractions from aerosol particles. Only regional surface–origin tracers with fractions greater than 20% are shown for better visibility. In intervals 1 and 3 enhanced fraction of the Western Pacific and in interval 2 of the South Asia tracer are found. Note that $H_2O$, $RH_i$ and nitrate are smoothed over 10 sec.



**Figure 9.** Frequency distribution of the locations where air parcels were traced back from the flight track of research Flight F02 on 6 August 2023 to the model BL and mean transport time for the entire flight (above 340 K) and time intervals 1–3. Reddish colours indicate mean transport times of a few days found for air from Bay of Bengal in time interval 1 and from Southeast Asia in time interval 1 and 3. Short transport times found in time interval 1 and 3 from the Western Pacific have a coincidence with the storm track of tropical cyclone Doksuri (C) (20–30 July 2023, for more details to cyclone category see Fig. 4).





### 4.3.2 Research flight F06 (16 August 2023): Typhoon air within the western outflow the anticyclone

Research flight F06 on 16 August 2023 conducted from Oberpfaffenhofen, Germany, to the Mediterranean area (in direction
of South Italy) reached the westward outflow of the Asian summer monsoon anticyclone which is mixed with air from the
Western Pacific (Fig. 10). The main source region of the outer edge of the ASMA is the Northern Western Pacific (Fig. 11).
At the southern turning point of research flight F06, a filament with high fractions of the Western Pacific tracer is reached. In
this filament indicated as interval 1 (in Fig. 11) the fractions from the Western Pacific tracer (up to 40%) are higher than from
the South Asia tracer (about 20%). Within interval 1 also the tropopause is enhanced or a second tropopause exists (indicating
Rossby wave breaking along the subtropical jet (Vogel et al., 2016)) compared to other flight segments. Therefore within
interval 1 the flight track was below the tropopause or between the first and second tropopause.

A comparison with chemical trace gases shows that in interval 1 the mixing ratios of $CH_2Cl_2$ and $CH_4$ are lower than in the
western part of the ASMA during research flight F02 (interval 2 in Fig. 8), but still higher than in the stratospheric background
at these altitudes. Low amounts of $O_3$ (below 100 ppbv) and enhanced amounts of $CH_2Br_2$ (higher than 0.8 pptv) indicate the
influence from marine sources in interval 1 (Fig. 11).

Dehydration occurs mostly in flight segments after interval 1 and sometimes also before interval 1 indicating that the air
parcels have passed the cold point tropopause. During interval 1 no dehydration was found even in the flight segment above
the 1st tropopause (but below the second tropopause) indicating moistening of the lower stratosphere in flight segments with
enhanced fractions of the Western Pacific. This indicates another origin and transport pathway of air masses in interval 1
compared to the adjacent flight segments.

The South Asia tracer has low fractions of about 20% in interval 1 indicating some outflow of the ASMA to the west.
However, during research flight F06, the low nitrate concentrations including interval 1 indicate that the flight was outside
the ATAL and that the air masses likely did not originate from the ATAL. In contrast, no enhanced levels of sodium chloride-,
TMA-, or MSA-containing particles were detected during Interval 1 (not shown), suggesting that marine-sourced particles may
have been removed during transport, possibly through washout processes.

The origin of the back-trajectories in interval 1 coincide with the storm track of tropical cyclone Khanun (26 July – 11 August
2023) (Fig. 12). The temporal evolution of the Western Pacific tracer at 370 K potential temperature (Fig. C3 in Appendix C)
shows the impact of tropical cyclone Khanun in particular on 1 and 2 August 2023. Subsequently, enhanced fractions of the
western Pacific tracer uplifted by tropical cyclone Khanun were transported anticyclonic around the ASMA's southern edge.
These air masses were measured during research flight F06 in interval 1 (Fig. 10).

Research flight F06 on 16 August 2023 demonstrates the impact of tropical cyclones in the western Pacific on the UTLS
over South Europe, transporting a relatively high amount of $CH_2Br_2$ and low $O_3$ to the UTLS over South Europe and most
likely further into the lower extra-tropical stratosphere within double tropopauses.



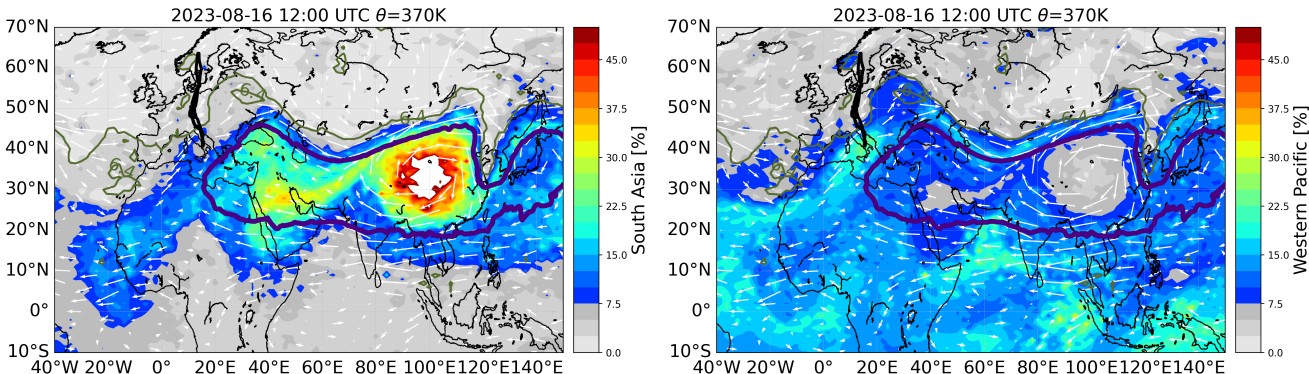

**Figure 10.** Research flight F06 on 16 August 2023 conducted from Oberpfaffenhofen, Germany, to the Mediterranean area (in direction of South Italy) reach the outer edge of the ASMA (South Asia surface–origin tracer at 370 K 12:00 UTC, left), which is dominated by air from the western Pacific (Western Pacific surface–origin tracer at 370 K 12:00 UTC, right). The surface–origin tracer distributions are based on a CLaMS simulation driven by ERA5. The calculated synoptic HALO flight track position at 12:00 UTC is indicated by the black line. To indicate the edge of the ASMA (indigo line), the boundary of the ASMA is calculated using the Montgomery streamfunction. An optimised background value gives the ASMA boundary (MSF = 356.6 m$^2$s$^{-2}$) for 6 August 2023 12:00 at 370 K using ERA5 reanalysis data based on the method by Kachula et al. (2025). The climatological isentropic transport barrier (PV = 6.4 PVU) derived by Kunz et al. (2015) for the Northern Hemisphere at 370 K during summer indicates the barrier between the tropical tropopause layer and the extra-tropical lower stratosphere (olive line). Further, horizontal winds are indicated by white arrows.







**Figure 11.** Surface–origin tracers interpolated along the flight track of research flight F06 on 16 August 2023, potential temperature, potential temperature at the tropopause as well as measurements of different chemical trace gases such as $CH_2Cl_2$, $CH_4$, $CH_2Br_2$, $O_3$, $H_2O$, relative humidity with respect to ice $RH_i$ as well as nitrate mass concentration from aerosol particles. Regional surface–origin tracers with fractions greater than 5% are shown. In interval 1 enhanced fraction of the South Asia and Western Pacific tracer are found; the dominant regional source region is the Northern Western Pacific (NWP). Note that $H_2O$, $RH_i$ and nitrate are smoothed over 10 sec. Along the $H_2O$ time series, flight segments are colour-coded in light-blue, when along the CLaMS back-trajectories possible dehydration is indicated.



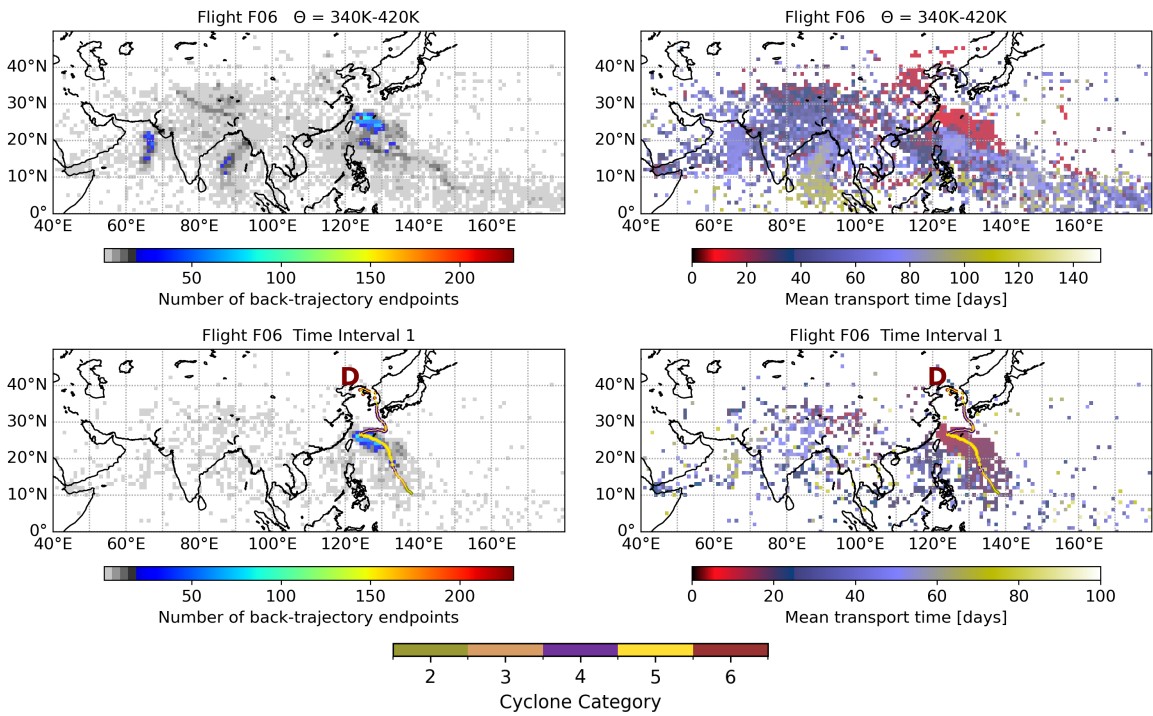

**Figure 12.** Frequency distribution of the locations where air parcels were traced back from the flight track of research Flight F06 on 16 August 2023 to the model BL and mean transport time for the entire flight (above 340 K) and time intervals 1. Reddish colours indicate mean transport times of a few days found for air from the Northern Western Pacific during time interval 1. Short transport time in time interval 1 coincidence with the storm track of tropical cyclone (D) Khanun (26 July – 11 August 2023, for more details to cyclone category see Fig. 4).





### 4.3.3 Research flight F08 (26/27 August 2023): The eastern outflow of the anticyclone

Research flight F08 on 26/27 August 2023 conducted from Anchorage, measured air in a filament over Alaska separated from the ASMA, which is well mixed with air from the western Pacific (Fig. 13). This filament is separated from the northern extratropical UTLS by the subtropical jet that constitutes a transport barrier on an isentropic level. Subsequent mixing of this air into the northern extratropical UTLS can occur (Vogel et al., 2016). Air masses measured in the eastern outflow of the ASMA are mainly from East China and the Western Pacific (ECH, NWP, TWP) (Fig. 14) in contrast to the outflow to the west that originates mainly from the Indian subcontinent (Fig. 7). Within this filament, the South Asia and Western Pacific tracer have similar fractions and vary in a similar way in contrast to flight F02, where the two tracers vary in an opposite way. This indicates that during the separation of the filament from the ASMA, air from inside the ASMA (high amounts of the South Asia tracer) was mixed with air located at its outer edge (high amounts of the Western Pacific tracer). The temporal evolution of the South Asia and the Western Pacific tracer at 370 K potential temperature from 19 to 27 August 2023 is shown in Fig. C4 (Appendix C) demonstrating the mixing of these two air masses.

The time series of the surface–origin tracers interpolated along the flight track of research flight F08 are also compared with measurements of several chemical trace gases (Fig. 15). The temporal variability along the flight track of $CH_2Cl_2$, $CH_4$ and $CH_2Br_2$ show a good overall agreement with the variability of the South Asia tracer and the Western Pacific tracer; vice versa $O_3$ shows opposite variations. This demonstrates that both polluted air from South Asia as well as marine air contribute to the chemical composition of the filament separated from the ASMA. Caused by the mixing of two air masses with different tropospheric origin, the chemical composition of the filaments separated at the eastern edge of the anticyclone is different from the chemical composition of the ASMA's air itself measured during research flight F02.

Three intervals 1–3 at different levels of potential temperature are marked in Fig. 15 highlighting enhanced nitrate mass concentrations in aerosol particles as an indicator of air from the ATAL (the small peak at 21:40 is not considered here, but yield similar results as for interval 1, however with a lower amount of air parcels). Similar as for F06 also here in F08, no enhanced levels of sodium chloride-, TMA-, or MSA-containing particles were detected during interval 1–3 (not shown), suggesting that marine-sourced particles may have been removed during transport to potential temperature levels above 360 K, possibly through washout processes.

In contrast to research flight F02 inside the western part of the ASMA, the flight segments of research flight F08 above $\sim$360 K were also above the local 1st tropopause, thus in the lower stratosphere (Fig. 15). In intervals 1, 2 and 3 (all three intervals are above the tropopause) also water vapour is enhanced compared to adjacent flight segments at the same level of potential temperature. Here, in intervals 1–3, cloud formation processes along the flight track can be excluded (RH$_i$ $\lesssim$10%). Findings from back-trajectory calculations show that in intervals 1, 2 and 3 dehydration does not occurs (only for a minor amount of trajectories), thus enhanced $H_2O$ is found within the filament separated from the ASMA. In particular in intervals 2 and 3, hydration is evident above 380 K potential temperature.

An accumulation of trajectory endpoints is found in South Asia as well as in the western Pacific in agreement with simulations using surface–origin tracers (Fig. 16). The origin of the back-trajectories in intervals 1, 2 and 3 coincide with the storm





track of several tropical cyclones (Fig. 16). The time intervals are on different levels of potential temperatures whereby the potential temperature is increasing from interval 1 to 3. The origin of the back-trajectories in the selected intervals coincide

with the storm track of tropical cyclones (B) Talim (13–18 July 2023), (C) Doksuri (20–30 July 2023), (D) Khanun (26 July – 11 August 2023), however for the highest potential temperature level at $\sim 400\,\mathrm{K}$ in interval 3, only Talim and Doksuri play a role. Long transport times ($\sim$40–60 days) from the model boundary layer to $400\,\mathrm{K}$ potential temperature compared to lower levels are found.

      Filaments separated at the eastern flank of the ASMA transport a mixture of polluted air from South Asia and marine air form

the western Pacific towards the lower extratropical UTLS. Thus, regarding the eastern outflow of the ASMA by the separation of filaments or eddy shedding, our findings demonstrate that within these filaments also contributions of air from the western Pacific have to be taken into account.



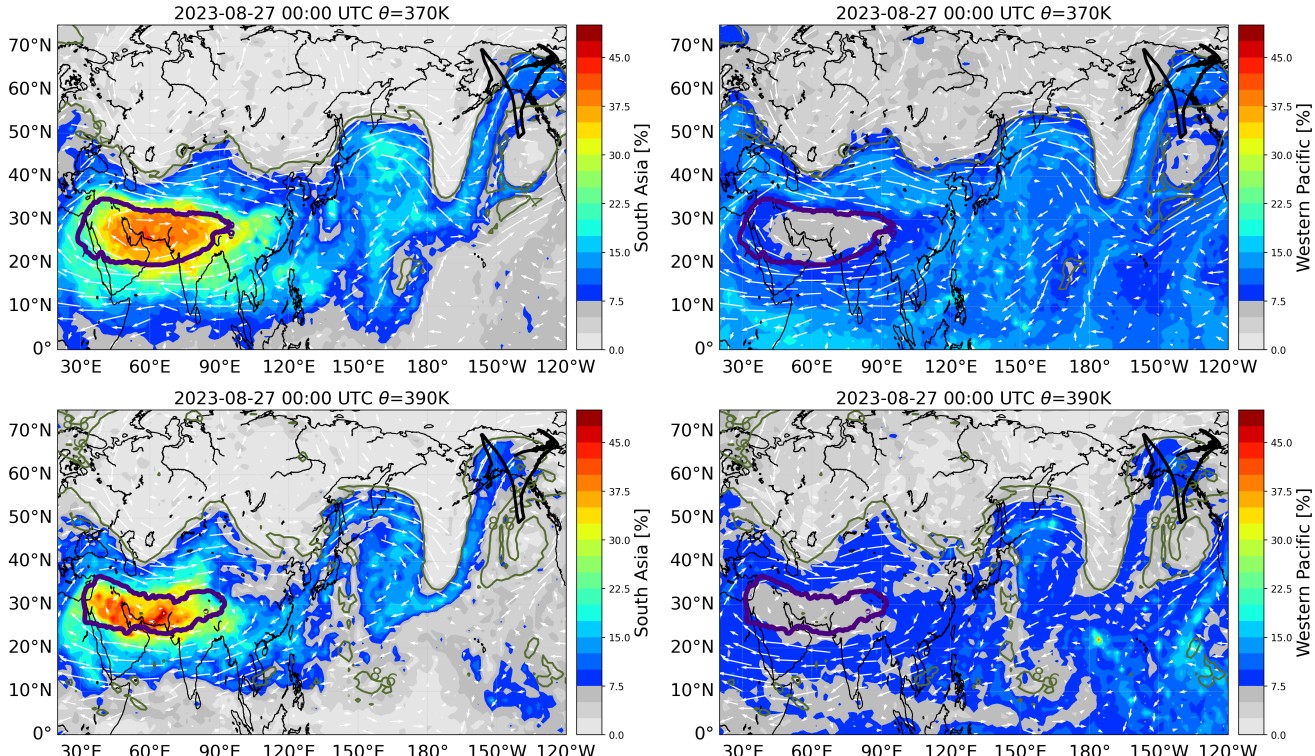

**Figure 13.** Research flight F08 on 26-27 August 2023 conducted from Anchorage probing a filament over Alaska separated from the ASMA (South Asia surface–origin tracer at 370 K and at 390 K 00:00 UTC, left), which is mixed with air from the western Pacific (Western Pacific surface–origin tracer at 370 K and 390 K 00:00 UTC, right). The surface–origin tracer distributions are based on a CLaMS simulation driven by ERA5. The calculated synoptic HALO flight track position at 27 August 2023 00:00 UTC is indicated by the black line. To indicate the edge of the ASMA (indigo line), the boundary of the ASMA is calculated using the Montgomery streamfunction. An optimised background value gives the ASMA boundary (MSF = 357.0 m$^2$s$^{-2}$ at 370 K and 367.6 m$^2$s$^{-2}$ at 390 K) for 27 August 2023 00:00 using ERA5 reanalysis data based on the method by Kachula et al. (2025). The climatological isentropic transport barrier (PV = 6.4 PVU at 370 K and 8.6 PVU at 390 K) derived by Kunz et al. (2015) for the Northern Hemisphere during summer (JJA) indicates the barrier between the tropical tropopause layer and the extra-tropical lower stratosphere (olive line). Further, horizontal winds are indicated by white arrows.



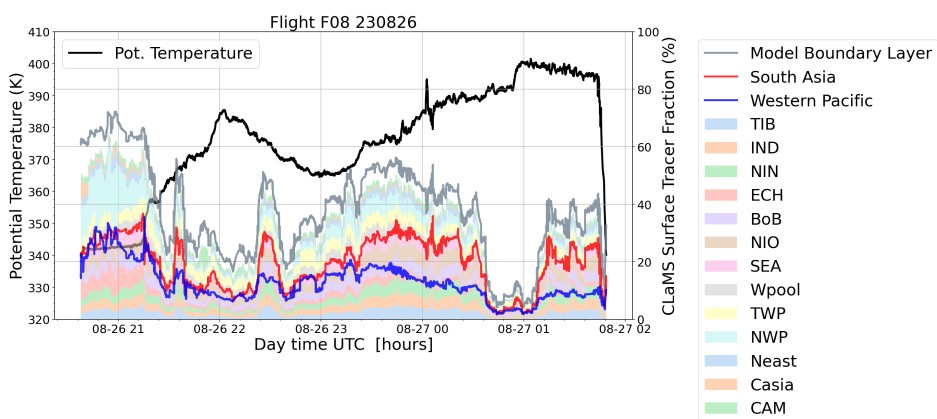

**Figure 14.** Surface–origin tracers for the model boundary layer, South Asia and the Western Pacific interpolated in space and time along the flight track of research flight F08 on 26 August 2023. In addition individual surface–origin tracers with fractions greater than 5% are shown. The main contributions are from China and the western Pacific (ECH, NWP, TWP).



**Figure 15.** As Fig. 11 but for research flight F08 on 26-27 August 2023. Regional surface–origin tracers with fractions greater than 10% are shown. Three time intervals (1-3) high-lighting enhanced nitrate mass concentrations in aerosol particles as an indicator of air from the ATAL as well as enhanced fraction of the South Asia tracer (indicating ASMA air) and the Western Pacific tracer are marked in grey.







**Figure 16.** Frequency distribution of the locations where air parcels were traced back from the flight track of research Flight F08 on 26-27 August 2023, to the model BL and mean transport time for the entire flight (above 340 K) and time intervals 1–3. An accumulation of trajectory endpoints is found in South Asia as well as in the western Pacific. The time intervals are on different levels of potential temperatures whereby the potential temperature is increasing from interval 1 to 3. The origin of the back-trajectories in the selected interval coincide with the storm track of tropical cyclones (B) Talim (13–18 July 2023), (C) Doksuri (20–30 July 2023), (D) Khanun (26 July – 11 August 2023), however for the highest potential temperature level in interval 3, only Talim and Doksuri play a role (for more details to cyclone category see Fig. 4).





## 5 Conclusions

We combined PHILEAS HALO aircraft measurements with global three-dimensional CLaMS simulations, including model
tracers of surface origin and with pure CLaMS back-trajectory calculations started along all aircraft flight tracks. Both three-dimensional simulations and back-trajectory calculations are driven by the high-resolution ERA5 reanalysis. Boundary source regions and transport times of air masses measured during the PHILEAS campaign were identified. Main source regions in South Asia and in the western Pacific (impacted by tropical cyclones) were found. Caused by the high-resolution of the ERA5 reanalysis data and the Lagrangian transport and irreversible mixing in CLaMS, a remarkable agreement between fine-scale
structures found in measured chemical tracers and in simulated surface–origin tracers along the flight tracks are found.

Our findings show that for the PHILEAS campaign, the South Asia tracer serves as a reliable proxy for polluted air originating from the Asian summer monsoon region. Additionally, the Western Pacific tracer is a useful marker for air masses of marine origin, and more specifically, an indicator of air uplifted into the UTLS by tropical cyclones in the western Pacific.

Regional surface–origin tracers, sub-regions of the South Asia or the Western Pacific tracer such as Northern Indian Subcon-
tinent (NIN), Indian Subcontinent (IND), Tibetan Plateau (TIB), Eastern China (ECH), Bay of Bengal (BoB), Indian Ocean (IND) or Northern Western Pacific (NWP) narrow the air mass origin of the PHILEAS measurements to smaller source regions.

Measured mixing ratios of $CH_2Cl_2$ from the HAGAR-V instrument depend strongly on the specific source region in Asia and 3 different branches with enhanced $CH_2Cl_2$ (higher than the background value at tropopause heights of ~50 pptv) and $CH_4$ larger than 1920 ppbv (as an indicator for ASMA air) are found. The highest $CH_2Cl_2$ mixing ratios ($200 - 300$ pptv)
are found from sources in China mixed with air from the Northern Western Pacific, however at altitudes below the ASMA ($\leq 360$ K). Further, CLaMS simulations indicate, that $CH_2Cl_2$ mixing ratios of $\sim 150$ pptv are a mixture of air from eastern China, Northern Indian Subcontinent and the Tropical Western Pacific; that is also valid for potential temperature levels below 360 K. $CH_2Cl_2$ mixing ratios of $\sim 100$ pptv, at altitude of the ASMA ($\geq$360 K) are attributed with air masses mainly from the Indian Subcontinent and Bay of Bengal.

Our case studies regarding three single research flights (F02, F06, and F08) are focused on the western part of the ASMA and its outflow to the west and east at potential temperature levels $\geq 360$ K. Measurements of pollutants and greenhouse gases such as $CH_2Cl_2$ and $CH_4$ as well as enhanced nitrate mass concentrations (a sign for particles from the ATAL) indicate sources in South Asia in agreement with CLaMS South Asia surface–origin tracer finding sources in the region of the ASMA. Our findings show that both the ASMA itself as well as tropical cyclones can enhance water vapour in the UTLS.

Measurements of ozone-poor and relatively enhanced $CH_2Br_2$ air that has mostly natural oceanic sources indicate sources in the Western Pacific in agreement with CLaMS Western Pacific surface–origin tracer as well as back-trajectory calculations indicating possible impact of tropical cyclones. Despite research flights F02, F06, and F08 were influenced by tropical cyclones and have marine sources, no significant particle fractions from sea salt and MSA as marine tracers were found. However enhanced TMA was found in the western part of the ASMA with enhanced influence from Northern India, in particular
Bengalen, with potentially large agricultural sources for TMA.



PHILEAS measurements demonstrate that the outer edge of the ASMA in 2023 is impacted by marine air from the western Pacific uplifted by tropical cyclones. Therefore, the chemical composition of the ASMA at its edge is highly inhomogeneous and depends on the interplay of tropical cyclones and the ASMA.

Subsequent mixing of polluted air from inside the ASMA and marine air (including enhanced natural $CH_2Br_2$) at its edge
occurs at the eastern flank of the anticyclone when filaments are separated from the main anticyclone. Therefore, the chemical composition of air in the eastern outflow of the anticyclone is different compared to the western part of the ASMA. Furthermore, air masses measured in the eastern outflow of the ASMA are mainly from East China and the Western Pacific in contrast to the outflow to the west that originates mainly from the Indian subcontinent.

The uplift of marine air by tropical cyclones and the subsequent transport impacted by the ASMA is a potential source of
water vapour as well as $CH_2Br_2$ (an ozone-depleting very short-lived substance) and low ozone contributing to the chemical composition of the UTLS. We emphasise that the intensity and duration of tropical cyclones has been increasing in recent decades (e.g. Emanuel, 2005; Mei and Xie, 2016; Knutson et al., 2020; Bhatia et al., 2022). Therefore, direct injections of marine air by rapid uplift in tropical cyclones to the outer edge of the ASMA will likely increase in the future.

*Code and data availability.* Observational data from the PHILEAS HALO mission are available via the HALO database (https://halo-
db.pa.op.dlr.de/). The ERA5 data used here are available from the ECMWF (https://www.copernicus.eu/en). The tropical cyclone tracks in the western Pacific are provided by the Japan Meteorological Agency and are available under https://www.jma.go.jp/jma/jma-eng/jma-center/ rsmc-hp-pub-eg/besttrack.html. The CLaMS code is available on a GitLab server at https://jugit.fz-juelich.de/clams/CLaMS. Results of the CLaMS simulations presented in this work are available from the corresponding author upon request.



# Appendix A: Additional tracer-tracer relations

## A1 $CH_4$–$CH_2Cl_2$ relations

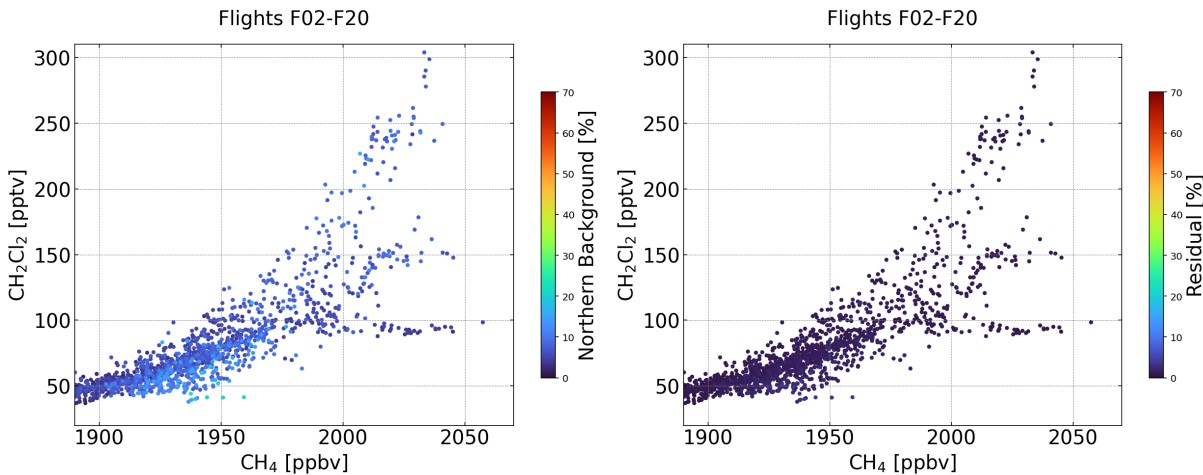

**Figure A1.** The same as Fig. 5, but colour-coded by CLaMS surface–origin tracers: Northern Background (CAM + Casia + NAM +NAO + NEA + NEP + NPO + TEP) and Residual (AMZ + AUS + NEZ + SAF + SAM + SAO + SEP + SIO + SPO + SWP + TAO).

## A2 $CH_4$–$H_2O$ relations indicating water vapour sources in South Asia and the western Pacific







**Figure A2.** The same as Fig. 5, but for surface–origin tracers contributing to the South Asia (IND, NIN, ECH, TIB, BoB, NIO, Neast, NAF) and Western Pacific tracer (Wpool, TWP, NWP, SEA).





**Figure A3.** $CH_4$–$H_2O$ relations for all scientific PHILEAS flights F02–F20 for measurements above 360 K, i.e. at altitudes of the ASMA colour-coded by the South Asia and Western Pacific tracer and potential temperature. In addition, also the scientific flights F02 (6 August 2023), F06 (16 August 2023) and F08 (26-27 August 2023) are shown.



## Appendix B: Bipolar mean spectrum of the TMA-containing particle type

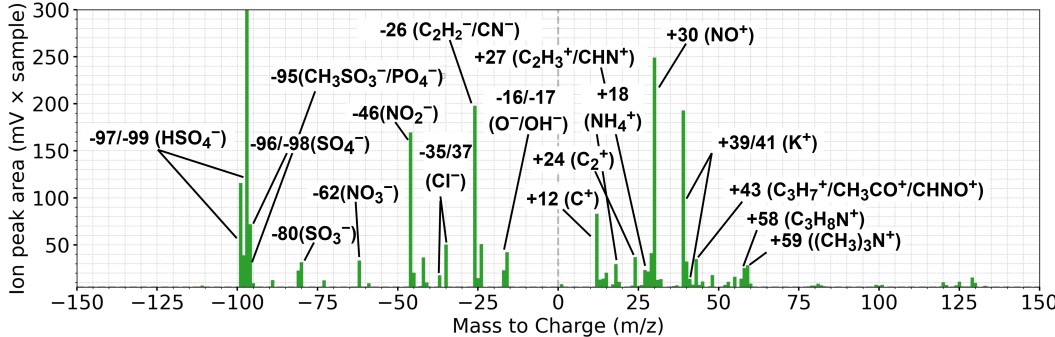

**Figure B1.** Bipolar mean spectrum of the TMA-containing particle type (average over 354 particles) measured with ERICA-LAMS during F02. The mean spectrum indicates that TMA in single particles is internally mixed with sulfate, nitrate, ammonium, potassium, chloride, MSA, and organic compounds. The signal intensity axis is limited to 300 mV, while the m/z -97 actually reaches up to 3000 mV.



## Appendix C: Temporal evolution of the South Asia and Western Pacific tracer

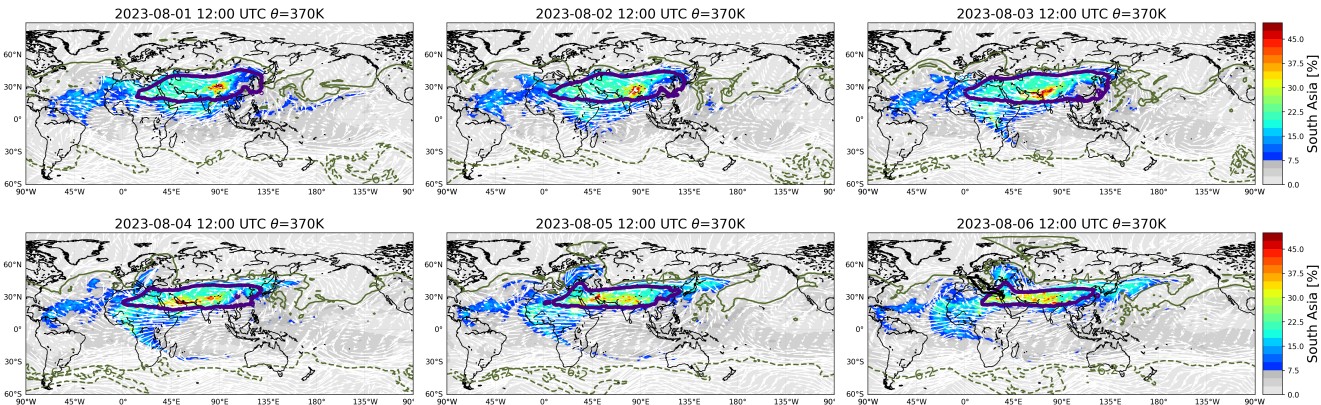

**Figure C1.** Temporal evolution of the South Asia tracer at 370 K potential temperature from 1 to 8 August 2023. On 6 August 2023 the calculated synoptic HALO flight track position at 12:00 UTC of research flight F02 is indicated (black line). The climatological isentropic transport barrier for the Northern (PV = 6.4 PVU; solid olive line) and Southern (PV = –6.2 PVU; dashed olive line) Hemisphere at 370 K during summer derived by Kunz et al. (2015) indicates the barrier between the tropical tropopause layer and the extra-tropical lower stratosphere.






**Figure C2.** Temporal evolution of the Western Pacific tracer at 370 K potential temperature from 15 to 31 July 2023. The impact of several tropical cyclones on the Western Pacific tracer is evident (reddish colours): Talim (B) (14–16 July), Doksuri (C) (23–27 July) and Khanun (D) (29 July – 2 August; see next figure for August). Subsequent enhanced fractions of the Western Pacific tracer uplifted by tropical cyclones were transported anticyclonic around the ASMA's southern edge. Thereafter, at the western flank of the ASMA over Africa or the Atlantic Ocean the air is transported eastwards by the subtropical jet and a belt of the Western Pacific tracer is developing between the northern edge of the ASMA and the subtropical jet. The climatological isentropic transport barriers (olive lines) are the same as in Fig. C1.







**Figure C3.** Same as Fig. C2, but from 1 to 18 August 2023. On 6 and 16 August 2023 the calculated synoptic HALO flight track position at 12:00 UTC of research flight F02 and F06, respectively, are indicated (black line).



**Figure C4.** Temporal evolution of the South Asia and the Western Pacific tracer at 370 K potential temperature from 19 to 27 August 2023. On 27 August 2023 (00:00 UTC; in contrast to all other panels that are shown on 12:00 UTC) the calculated synoptic HALO flight track position of research flight F08 is indicated (black line).



*Author contributions.* BV developed the concept for this study and performed the CLaMS simulations with contributions of JC. VL, JS,
RvL, MCV, FE, FK, SB, AD, OE, SM, JS, PH, LO, FW, CR, and AZ conducted the HALO PHILEAS measurements and the subsequent data analysis of these observations. MR, PH and CR coordinated the PHILEAS campaign including the scientific flight planning. OK provided the ASMA boundaries. BV wrote the paper with contributions of all authors. The interpretation of the results was discussed with all coauthors.

*Competing interests.* At least one of the (co-)authors is a member of the editorial board of Atmospheric Chemistry and Physics.

*Acknowledgements.* We are grateful to all colleagues that were involved in the PHILEAS project, in particular applying and coordinating
the project, planning the research flights including the provision of forecast products as well as preparing and conducting measurements onboard HALO and the subsequent data analysis. In particular we would like to thank Florian Obersteiner (KIT, Karlsruhe, Germany) for the FAIRO data, Daniel Kunkel (University Mainz, Germany) and Johannes Schneider (MPI, Mainz, Germany) for further discussion of the manuscript. We thank the European Centre for Medium-Range Weather Forecasts (ECMWF) for providing the the ERA5 reanalyses and the Jülich Supercomputing Centre (JSC; Research Centre Jülich, Germany) for the computing time on the supercomputer JUWELS (project
CLaMS-ESM) and for the storage resources. Finally, we acknowledge Matthias Riße and Nicole Thomas very much for their support on a variety of technical issues related to this work. We also thank ChatGPT for assisting in language refinement and python programming. The presented work was partly funded by the German Science Foundation (Deutsche Forschungsgemeinschaft, DFG) as part of the HALO Priority Program SPP 1294 (VO 1276/7-1, HO-4225/17-1 HO-4225/19-1, KO 6470/1-1), the NSFC–DFG project ATALTrack (VO 1276/6-1 and BO 1829/12-1) and the TRR 301 TPChange project (Project-ID 428312742).



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
