# Peer review of "Continental and marine source regions contributing to the outflow of the Asian summer monsoon anticyclone during the PHILEAS campaign in summer 2023"

_EGUsphere, 2025_

## Referee Comment (RC1)

**Review of "Continental and marine source regions contributing to the outflow of the Asian summer monsoon anticyclone during the PHILEAS campaign in summer 2023" by Vogel et al.**

**Summary:** This paper integrates Lagrangian transport simulations with airborne in situ observations from the 2023 PHILEAS campaign to derive insights about the source regions contributing to the composition of the Asian summer monsoon UTLS anticyclone. Three-dimensional tracer simulations as well as backward trajectory calculations from the PHILEAS flight tracks are used. The authors highlight three case studies which show the important role of marine sources (Pacific tropical cyclones in particular) that contribute to the air masses sampled by the HALO research aircraft.

**Overall Thoughts:** This is a well-written paper which underscores the important contribution of western Pacific air masses to the composition of the Asian monsoon UTLS region. This contribution has been identified in the past, but it has remained somewhat unappreciated, making this an important contribution. I believe this work should be published after the authors take into account my mostly minor remarks below.

**Recommendation:** Minor Revision

**General Remarks:**

- I am a bit concerned about the authors choice of "South Asia" as the terminology to describe everything from northern Africa to eastern China, particularly for the reasons below:
    - 1. The northern Africa (NAF) domain seems to go all the way to the Atlantic coast (Figure 2), which is very far from Asia. If the NAF domain makes an important contribution (I didn't personally get this impression, although line 176 suggests that it was added deliberately for this manuscript), then the authors should add a clarifying remark about why it was included as part of "South Asia". If not, then I suggest redefining "South Asia" to exclude the NAF region for clarity.
    - 2. Recent results from the ACCLIP (2022) campaign have illuminated the important contribution of both South Asia *and* East Asia to the composition of eastward-transported ASM air masses (see for example Pan et al., 2025; https://doi.org/10.1029/2025JD044417, and several references therein). Given this, I question whether it's appropriate for these regions to be combined and referred to as "South Asia" in this work without explanation. In particular, the prominent source of dichloromethane is "East Asia" (i.e.,

China) but the authors state that elevated concentrations indicate a source from "South Asia" (lines 9 and 262 at least) which I find misleading. I don't suggest that the analysis be totally redone, but I do think the authors should acknowledge recent literature that finds both these regions to be important, and provide a justification for why they are not considered individually in the current work.

- o 3. Related to the above remark, I am actually a bit perplexed that CLaMS results don't suggest a stronger contribution from "East Asia" (e.g., Figure 3). Recent work by Jesswein et al (2025; https://doi.org/10.5194/acp-25-8107-2025) traced PHILEAS measurements to sources over East Asia. One of the flights (F08) is common between these two studies, but the results seem quite different. I understand these studies use different modeling approaches, but I think it could be insightful for the authors to provide some explanation for these differing results.

- The argument that oceanic air mass influence is important for ASM UTLS composition is expertly made, however I don't expect the oceanic contribution to be important in all cases. I think that the authors should acknowledge that the research flights emphasized in this work are chosen especially because they highlight tropical cyclone influence to make the point that it can play an important role. The authors could consider additional figures in the appendix showing the top panels of figures 9, 12 and 16 but for all 20 flights if they want to conclude that this contribution is routine, but otherwise it should be clear that the conclusions are only valid for the chosen cases. More comprehensive modeling analysis over a broader region and time period (not just targeting aircraft measurements) might still be needed to make truly general statements about oceanic air mass contribution, though I expect this is outside the scope of the current work.

**Technical Remarks and Typos:**

- There are several places in the manuscript that the authors use "western" and "eastern" instead of "westward" and "eastward" when describing outflow from the ASM. I recommend using the latter terminology. Some locations I found were lines 5, 8, 76, 445, 486, and the Section 4.3.2 title, though there may be others.
- Lines 6-7: This sentence is a bit short and general. Perhaps: "The current work integrates PHILEAS aircraft in situ measurements with output from Lagrangian transport simulations to..." ?

- Line 9: maybe specify nitrate aerosol, since that seems to be the emphasis.
- Line 28: I would remove "(e.g., Alaska)", I don't believe the ASM air mass was sampled over Alaska, but rather that Alaska was used as a base to reach the ASM air mass to its south (over the North Pacific).
- Line 31: remove extra "the"
- Line 41: I would remove the "(in particular typhoons)" remark, or change to "including typhoons". All typhoons are tropical cyclones, and the latter should be considered broadly important for transport even if they're not at typhoon status.
- Line 54: aerosol backscatter is not a trace gas, but the sentence structure suggests that it is.
- Line 75: "ASMA" typo
- Line 81: I suggest "Anchorage, Alaska (USA)" since the country "Germany" is in parentheses before this.
- I was a bit confused in a few spots about how F02-F20 was called "20 flights". Numerically this should only be 19 flights. I see that there are two days that have "subflights" (a/b) and F03 also seems to be excluded as well. I suggest adding a short clarification about this, and also mention why F03 was excluded (as F01 is mentioned in the Figure 1 caption).
- Line 88: "database" ?
- Line 136: The ERA5 data were retrieved on a regular horizontal grid. Were they also retrieved with a degraded vertical resolution compared to the 137 native levels? That would be a good to clarify as well.
- Line 155: I'm not totally comfortable with ~2-3 km above the surface being referred to as the "boundary layer". I'm guessing that may be true in some places, but that altitude may be the lowermost free troposphere in others. I wonder if the authors would be comfortable renaming this layer the "lower troposphere" throughout.
- Line 184: 20 "science" flights?
- Related to the first general remark above, I see that the north Indian Ocean (NIO) region is also included in the analysis which spans all the way into the southern hemisphere (thus far from Asia). I am left wondering whether most of the contribution from "South Asia" comes from the land regions (where anthropogenic pollution would be found in reality) and regions like the NIO and BoB are minor.
- Line 200-209: There is some redundant text in this area. The sentence that ends with "remain in the stratosphere afterwards" is repeated twice, for example.

- Figure 3 and others: since there are ~30,000 backward trajectories per flight (totaling ~600,000 for 20 flights), I'm struggling to believe that the colorbars show an absolute number of backward trajectory endpoints. Is there a scalar applied? Perhaps I just need to be reassured by the authors that there are enough pixels to reach that large a number.
- Figure 4 and others: I don't personally like the Cyclone Category colorbar used on several figures, ranging from 2-6. For one, there is already a numerical scale for typhoon / hurricane categories, which could lead to confusion. Moreover, number 6 is numerically highest but refers to extra-tropical cyclones which are typically weak. I suggest assigning acronyms (perhaps TD, TS, STS, Typ, ExTC) to the colorbar labeling to help with reader comprehension.
- Figure 4 and others: I would also suggest putting the colored dots (for the cyclone category) on top of the yellow cyclone track lines, as they are currently hard to see.
- Line 279: I suggest replacing "it turns out" with "reveals".
- Line 302: I suggest "potential temperature levels".
- Line 322: remove extra "are"
- Line 342: Replace "relative" with "relatively".
- Line 349: "suggest" instead of "yield"?
- Line 355: "typhoon classification" instead?
- Line 357: I suggest "large fraction" instead of "high fraction".
- There are several places the authors refer to a "calculated synoptic flight track position". Why is this not just simply the "flight track" using output provided by onboard instrumentation?
- Figure 6 caption: I suggest specifying that horizontal winds are from ERA5 (if true).
- Figure 7 and others: The black line is labeled as "potential temperature" in the legend, but the y-axis has the same label. Perhaps label the black line as the "flight track" or "HALO aircraft" in the legend instead.
- Line 376: Why not call this region the "Northwestern Pacific" for simplicity?
- Line 399: Replace "anticyclonic" with "anticyclonically".
- Line 402: I suggest "large amount" instead of "high amount". The latter can refer to either altitude or concentration.
- Figure 10 and 13: I suggest adding simplicity to these captions with "As in Figure 6 but for flights on…"
- Line 411: I think that rewriting this sentence to use "positively/negatively correlated" would sound more scientific.
- Line 431: I suggest "at the same potential temperature".
- Line 433: typo, use "occur"
- Line 444: typo, use "from"

- Line 446: remove "also" or move it to after "Pacific".
- Figure 15a: I am a bit confused at how small the ECH TWP and NWP contributions are compared to the red and blue lines for the aggregated contributions. Are we sure these are the important source regions here, or is there some issue with the plot?
- Line 455: redundant "found", I suggest ending the sentence with "identified".
- Line 468: I suggest rewriting the sentence to not imply that ">360K" is an altitude.
- Line 470: I gathered that the analysis was focused on both the western and eastern parts of the ASMA, but the text mentions only the western part.
- Line 487: The concept of Eastern China appears here, but all through the manuscript this was hidden behind the veil of a "South Asia" label (see the first general remark).
- It seems to me that Figure A2 belongs under the Appendix A1 header.
- Figure C1 caption should read "1 to 6 August".
- Figure C2 caption line 4: "anticyclonic" should be "anticyclonically".
- Line 516: I'm not sure if it's more appropriate to "thank" or "acknowledge" artificial intelligence, the authors can decide :)
- Note that the Pan et al. (2025) paper (https://doi.org/10.1029/2025JD044417) was just published, so this reference should be updated accordingly.

---

## Referee Comment (RC2)

**Reviewer Report**

Manuscript Title: "Continental and marine source regions contributing to the outflow of the Asian summer monsoon anticyclone during the PHILEAS campaign in summer 2023"

Authors: Vogel et al.

**Reviewer Comments on the Introduction (Sections 1 and 2)**

General assessment

The Introduction (Sect. 1) and the campaign and instrumentation overview (Sect. 2) together provide a thorough and well-referenced foundation for the study. The authors successfully place the PHILEAS campaign in the context of the Asian Summer Monsoon Anticyclone (ASMA), associated transport pathways, and recent aircraft and balloon campaigns. Section 2, in particular, offers a detailed and technically sound description of the HALO measurements and demonstrates the high quality and breadth of the observational data set.

Major comments

1. Structure and narrative flow of the Introduction (Sect. 1): The Introduction is scientifically comprehensive but very dense, with long paragraphs covering multiple themes. Rather than adding subheadings, which may disrupt the flow of a standard ACP Introduction, the text would benefit from clearer thematic paragraph separation, for example:

- ASMA formation, vertical transport, and confinement within the anticyclone.

- Export pathways via filament shedding and eddies to the extratropics.

- The role of tropical cyclones (TCs) in modifying UTLS composition.

- Previous observational campaigns and the specific contribution of PHILEAS.

Such restructuring would improve readability and help guide the reader toward the study's main objectives.

In addition, a schematic summarizing transport pathways (South East Asia, Western Pacific) and the location of the campaigns would be helpful.

Also, 2023 was a very active Pyro Cb season in N. America (Peterson et al., 2025). It would be interesting to understand how it may have influenced PHLEAS measurements.

2. Clearer articulation of novelty and the role of PHILEAS (Sect. 1): Several recent campaigns are mentioned (StratoClim, TACTS/ESMVal, WISE, ACCLIP), but the unique contribution of PHILEAS is not emphasized early enough. The authors are encouraged to more explicitly state what PHILEAS adds beyond these efforts. In particular, PHILEAS provides a valuable dual-flank perspective, sampling both:

- the western part of the ASMA and its westward outflow (Mediterranean region), and

- the eastern flank and Pacific outflow (via flights from Alaska).

Highlighting this earlier would strengthen the motivation for the study.

3. Instrument calibration and measurement stability (Sect. 2): Instrument accuracy and uncertainty are reported for FAIRO and UMAQS, which is helpful. To further support data quality in the UTLS, the authors are encouraged to briefly mention:

- the frequency of in-flight calibrations, and/or

- observed instrumental stability or drift over the 20-flight campaign.

This information could be added succinctly without expanding the section substantially.

4. Rationale for trace gas selection (Sect. 2): While $CH_2Cl_2$ and $CH_2Br_2$ are highlighted as key species from HAGAR-V, the instrument measures a broader suite of tracers. A short sentence explaining why these short-lived halocarbons were prioritized (e.g., sensitivity to rapid uplift from polluted or marine boundary layers) would improve the scientific narrative and link the measurements more clearly to the study objectives. In this case, using CO as a tracer of deep convection over polluted areas would be useful.

5. Scientific role of transfer flights (Sect. 2): Transfer flights (F07a, F07b, F19) are included in the data set, but their scientific role is not clearly stated. It would be useful to clarify whether these flights are used to characterize background extratropical UTLS or stratospheric conditions, thereby providing a reference for ASMA-influenced air masses.

Minor comments and technical corrections

1. Typo (Sect. 1): "into the the tropical tropopause layer" → remove the duplicated "the."

2. Consistency (Line 31): Use "westward" instead of "westwards" to match "eastward."

3. Terminology (Line 45): Consider rephrasing "ozone-poor and aerosol-poor marine air" as "ozone- and aerosol-depleted marine air."

4. Line 47: You may add the influence of typhoons on cirrus cloud formation through stratospheric hydration and waves Pandit et al., 2024).

4. Acronym use (Sect. 1): After defining ASMA, please use the acronym consistently.

5. Conceptual clarity (Sect. 1): The term "flushed" is well known but could be briefly linked to isentropic transport for physical clarity.

6. Potential temperature phrasing (Sect. 2): Rephrase "~14.5 km (~410 K)" as "up to ~14.5 km, corresponding to potential temperatures up to ~410 K."

7. FISH correction (Sect. 2): Please indicate the typical magnitude of the gas-phase water correction in ice clouds to give context for measurement sensitivity.

8. Figure 1 clarity (Sect. 2): Ensure the final figure is not overly cluttered and that time axes are clearly labeled in UTC, given the mix of Alaska and Germany flight segments.

9. Line 50. A reference is missing.

**Reviewer comments on Section 3: Lagrangian transport simulations**

General Comments: The authors provide a robust description of the CLaMS model setup, the use of ERA5 reanalysis, and the implementation of surface-origin tracers. The distinction between the 3D Eulerian-Lagrangian simulations (including mixing) and the pure back-trajectory calculations is well-maintained. The use of specific tracer sums to define "South Asia" and "Western Pacific" origins is a powerful diagnostic for interpreting the PHILEAS measurements.

Major Comments

1. Resolved vs. Unresolved Convection: The authors explicitly state (Line 146) that no additional convective parameterization is used and that unresolved small-scale convection is not considered; in other words, tracer fractions represent a "lower limit." Given that the ASMA is driven by deep convection, the authors should briefly discuss the implications of this underestimation. Specifically, how might the omission of sub-grid scale convection affect the calculated "age of air" or the timing of the "flushing" of the stratosphere?

2. Definition of the "South Asia" Tracer: The South Asia tracer (Line 169) includes a broad range of regions, including "Northern Africa (NAF)" and "Near East (Neast)." While the authors note these contribute only small fractions, the NAF region is often associated with mineral dust and different chemical signatures than the anthropogenic-heavy IND/ECH regions. A brief justification or a sensitivity note on why these regions are grouped into the "South Asia" monsoon proxy would be beneficial.

3. Dehydration Methodology (Section 3.2): The criteria for identifying dehydrated air masses (Lines 206-209) are quite specific (e.g., 80% of the time below the tropopause before $H_2O_{sat,ice,min}$). Please clarify the sensitivity of the results to the "80%" threshold. Additionally, the use of a simple minimum saturation mixing ratio ($H_2O_{sat,ice,min}$) is a first-order approximation. Does this approach account for the fact that ERA5 temperatures may have a cold/warm bias at the tropical tropopause, which significantly impacts $H_2O_{sat}$?

4. Mixing vs. Trajectory Consistency: The 3D simulations include "irreversible mixing applied every 24 h," whereas the back-trajectories (Section 3.2) appear to be purely advective (kinematic/diabatic). The authors should comment on how the lack of mixing in the backward trajectories might lead to discrepancies when compared to the 3D forward-modelled tracer distributions at the flight track.

Minor Comments and Technical Corrections

1. Line 139 (Technical detail): The mention of MPI and shared memory storage is more of a technical implementation detail. While interesting, it could be shortened or moved to an appendix/supplement if space is a concern, as it doesn't directly affect the scientific interpretation.

2. Line 183: The section title "CLaMS back-trajectory calculations" might be more descriptive as "Back-trajectory analysis and dehydration criteria."

**Reviewer Comments on Section 4: Results**

1. Threshold Definition: The text states "measurements above about 360 K" (line 225), but Figure 3's bottom panel specifies "360K-420K." This discrepancy risks ambiguity. The manuscript should explicitly align the text with the figure's precise range (e.g., "measurements above 360 K, corresponding to the ASMA's upper convective outflow layer").

2. Cyclone Category Interpretation: The text claims cyclones (B) Talim, (C) Doksuri, (G) Saola, and (I) Haikui "were all categorized as a typhoon" (line 235), but Figure 4's legend defines "Typhoon" as category 5. Verify if all four cyclones are indeed category 5 (e.g., via JMA data) and correct the text if inconsistent.

3. "overlayed" → "overlaid" (line 230).

4. "relations relations" → "relations" (line 245).

5. Cyclone Impact Mechanism: The claim that cyclones "rapidly uplift polluted boundary layer air" (line 237) requires supporting evidence. While Figure 4 shows spatial overlap, the manuscript should link cyclone timing to flight dates (e.g., "Cyclone Khanun (D) impacted F08 on 26 July 2023, coinciding with elevated $CH_2Cl_2$ mixing ratios").

6. Reference Accuracy: "Gettelman and de Forster, 2002" (line 223) likely refers to Gettelman and Forster (2002), a standard citation for convective outflow heights. Correct the reference to avoid confusion.

7. Tracer-Trace Gas Linkage: Section 4.2 states the goal is to "demonstrate different chemical compositions" between source regions, but does not present results. Clarify whether tracer-tracer plots (e.g., $CH_4$ vs. $CH_2Cl_2$) are included in subsequent sections or if this section is purely methodological.

8. The text notes "frequent occurrence of strong tropical cyclones" (line 226) but does not quantify their contribution to total back-trajectory endpoints. Add a statistic (e.g., "30% of ASMA air masses originated from cyclone-impacted regions") to strengthen the claim.

9. Line 255: CH4 Threshold Ambiguity: The text states "air masses with CH4 mixing ratios exceeding a certain threshold," but does not explicitly define the threshold used in this study (e.g., 1850 ppbv vs. 1920 ppbv). While prior studies are cited, the manuscript should clarify whether a specific threshold was applied here or if the analysis focuses on all high-CH4 air masses (>2000 ppbv, as implied in the text). This is critical for reproducibility.

10. Line 271: The text describes "three distinct branches" but does not explain how these branches were objectively identified (e.g., statistical clustering, visual inspection). A brief methodological note (e.g., "branches were identified via k-means clustering with k=3") would strengthen the analysis.

11. Line 287: The claim that region 4 "indicates marine air from the western Pacific" relies on CH2Cl2 matching "northern hemispheric background values." However, background CH2Cl2 could also originate from other regions (e.g., North America). The manuscript should:

- Cite specific background data to support this conclusion.

- Acknowledge that CH2Cl2 alone cannot definitively distinguish Western Pacific marine air without corroborating tracers (e.g., CO, O3).

12. The text attributes high CH2Cl2 to South Asia but does not address potential contributions from other regions (e.g., Southeast Asia, East Asia). While Appendix A2 is referenced, the main text should briefly summarize key findings (e.g., "branch 1 is dominated by eastern China, while branch 3 is linked to the Indian Subcontinent and Bay of Bengal").

13. Line 288:  The text states "other parts of the world... have only a minor impact" but relies on Appendix A1 for evidence. The main text should summarize key findings (e.g., "Appendix A1 confirms northern background and residual surface tracers contribute <5% to PHILEAS measurements").

14. Line 307: The text states the ASMA boundary is calculated using the Montgomery streamfunction (Kachula et al., 2025), but the figure caption specifies a Montgomery streamfunction value (MSF = 357.3 m² s⁻²). Clarify whether the method relies on a fixed MSF threshold or a dynamic calculation. The 2025 reference is likely a typo (2024 or earlier); verify the correct citation.

15. Line 313: The text claims "opposite variations" between South Asia and Western Pacific tracers (Fig. 7), but Figure 7's color scale for "Western Pacific" (blue) and "South Asia" (red) shows overlapping trends in some segments. Quantify the degree of separation (e.g., correlation coefficients) to strengthen this claim.

16. The text links TMA to "marine sources and/or agricultural activities," but Figure 8f shows that TMA fractions are highest in interval 2 (South Asia-dominated) and 3 (Western Pacific-dominated). Explicitly test whether TMA correlates with both marine ($CH_2Br_2$) and anthropogenic ($CH_2Cl_2$) tracers to validate dual-source hypotheses.

17. 29. Line 362: The text asserts air masses in intervals 1 and 3 were "uplifted by tropical cyclone Doksuri" (Fig. 9), but Figure 9's "Cyclone Category" legend (bottom-right) shows only 2–3 categories. Clarify how cyclone intensity (e.g., typhoon vs. tropical storm) directly links to tracer uplift.

18. The authors use "particulate nitrate" as an ATAL indicator but do not quantify how nitrate mass concentrations in interval 2 (Fig. 8f) compare to established ATAL thresholds (e.g., >100 ng/m³). Provide context for this metric.

19. The text references the ASMA boundary calculation using the Montgomery streamfunction (MSF = 357.0 m² s⁻² at 370 K) in Figure 13's caption, but the main text does not explicitly define this threshold or its basis. Clarify whether this value is derived from a fixed climatological standard or dynamically optimized for the 2023 campaign.

20. Line 410:  The text states that "South Asia and Western Pacific tracer fractions have similar values" in interval 1, but Figure 15a shows overlapping contributions from multiple regions (e.g., ECH, NWP, TWP). Specify the minimum fraction required to classify air as

"South Asia-dominated" or "Western Pacific-dominated" and justify why 10% (as in Figure 15's caption) is significant.

21. Line 440: The text links interval 1–3 air to tropical cyclones (Talim, Doksuri, Khanun), but Figure 16's "Cyclone Category" legend shows only 2–3 categories. Quantify how cyclone intensity (e.g., typhoon vs. tropical storm) correlates with tracer uplift (e.g., Fig. C4 in Appendix C) to strengthen causality.

22. Line 418: The text states $CH_2Cl_2$, $CH_4$, and $CH_2Br_2$ "show a good overall agreement" with tracers, but Figure 15a–d shows asynchronous peaks (e.g., $CH_2Br_2$ peaks at 21:40 while tracers peak at 22:00). Quantify correlations (e.g., $R^2$ values) to validate the claim of "good agreement."

23. Line 427: The text suggests "marine-sourced particles may have been removed via washout," but no evidence of precipitation along back-trajectories is provided.

24. Line 444: The phrase "marine air form" contains a grammatical error ("form" should be "from"). Correct such errors and replace repetitive terms (e.g., "Western Pacific tracer" → "WP tracer" after first use).

25. Figure 15's caption states "flight segments are colour-coded in light-blue when dehydration is possible," but the text does not reference this. Explicitly link dehydration timing to the color-coding in the main text.

**Reviewer Comments on Section 5: Conclusions**

The Conclusions section provides a comprehensive synthesis of the PHILEAS campaign results and effectively highlights the combined roles of the Asian Summer Monsoon Anticyclone (ASMA) and tropical cyclones in shaping the chemical composition of the UTLS. The integration of HALO aircraft observations with CLaMS three-dimensional simulations and back-trajectory analyses is a clear strength of the study and supports many of the overarching interpretations. Nevertheless, several aspects of the Conclusions would benefit from clarification, moderation of claims, or additional contextualization to improve scientific rigor and clarity.

1. The Conclusions state that the South Asia tracer is a "reliable proxy" for polluted air and that the Western Pacific tracer is a "useful marker" for marine air uplifted by tropical cyclones. While these statements are plausible, they remain qualitative. It would strengthen the Conclusions if the authors briefly clarified what criteria underpin these characterizations (e.g., degree of correlation with measured tracers, consistency across flights, or robustness across potential temperature ranges). In addition, the manuscript refers to a background $CH_2Cl_2$ value of ~50 pptv at tropopause levels; a short justification or reference for this background level would help contextualize the reported enhancements.

3. The reported dependence of $CH_2Cl_2$ mixing ratios on both altitude and source region is a key result. For example, the highest values (200–300 pptv) are found below the ASMA (≤ 360 K), while lower values (~100 pptv) dominate at ASMA altitudes (≥ 360 K). The Conclusions would benefit from a short discussion of the physical or chemical processes that may explain this vertical structure, such as differences in convective efficiency, dilution during ascent, or chemical lifetime effects.

4. The Conclusions note the occurrence of ozone-poor air with enhanced $CH_2Br_2$ in the Western Pacific and attribute this to marine influence and possible cyclone-driven uplift. While this interpretation is reasonable, the relationship between ozone and $CH_2Br_2$ is not explicitly described. Indicating whether a clear (anti-)correlation is observed, even qualitatively, would strengthen the internal coherence of this argument

5. Enhanced trimethylamine (TMA) in the western part of the ASMA is attributed to agricultural sources in Northern India and the Bengal region. This is an interesting and potentially important finding, but the evidence supporting this attribution is not discussed in the Conclusions. A brief reference to prior studies or an explicit statement that this interpretation is tentative would make the conclusion more balanced.

6.The role of tropical cyclones in uplifting marine air into the UTLS is emphasized repeatedly and is central to the study's narrative. However, the Conclusions do not quantify the extent of this influence (e.g., the fraction of air masses at the ASMA edge linked to cyclone activity). Even an approximate or qualitative estimate would help readers assess the relative importance of cyclones compared to other transport pathways.

7. The manuscript highlights increasing intensity and duration of tropical cyclones in recent decades and suggests that direct injections of marine air into the UTLS will likely increase in the future. While this perspective is relevant, the connection between these long-term trends and the specific chemical impacts documented in this study could be articulated more explicitly to avoid the impression of speculation beyond the presented results.

8. The Conclusions correctly point out that $CH_2Br_2$ is an ozone-depleting very short-lived substance and that its transport into the UTLS may have implications for stratospheric ozone. A brief indication of the potential magnitude or relevance of this effect (even in qualitative terms) would help place this result in a broader atmospheric chemistry context.

9. Some redundancy is present, particularly regarding the role of tropical cyclones in uplifting marine air and influencing the ASMA edge. Condensing these statements could improve readability without weakening the main message.

Finally, the Conclusions do not explicitly address limitations of the study or outline directions for future research. Including a short statement on key uncertainties (e.g., trajectory limitations, sensitivity to reanalysis data, or tracer interpretation) and possible next steps would provide a more balanced and forward-looking conclusion.